# mTOR inhibition attenuates chemosensitivity through the induction of chemotherapy resistant persisters

Yuanhui Liu[1,2], Nancy G. Azizian[1,2], Delaney K. Sullivan [3] & Yulin Li [1,2] ✉

Chemotherapy can eradicate a majority of cancer cells. However, a small population of tumor cells often survives drug treatments through genetic and/or non-genetic mechanisms, leading to tumor recurrence. Here we report a reversible chemoresistance phenotype regulated by the mTOR pathway. Through a genome-wide CRISPR knockout library screen in pancreatic cancer cells treated with chemotherapeutic agents, we have identified the mTOR pathway as a prominent determinant of chemosensitivity. Pharmacological suppression of mTOR activity in cancer cells from diverse tissue origins leads to the persistence of a reversibly resistant population, which is otherwise eliminated by chemotherapeutic agents. Conversely, activation of the mTOR pathway increases chemosensitivity in vitro and in vivo and predicts better survival among various human cancers. Persister cells display a senescence phenotype. Inhibition of mTOR does not induce cellular senescence per se, but rather promotes the survival of senescent cells through regulation of autophagy and G2/M cell cycle arrest, as revealed by a small-molecule chemical library screen. Thus, mTOR plays a causal yet paradoxical role in regulating chemotherapeutic response; inhibition of the mTOR pathway, while suppressing tumor expansion, facilitates the development of a reversible drug-tolerant senescence state.

Genotoxic damage-inducing chemotherapy is the mainstay treatment for most cancers. Effective chemotherapeutic agents often kill a large fraction of tumor cells, while sparing a small surviving population, which can resurface as future tumor recurrence. Acquired therapeutic resistance occurs due to genetic and/or non-genetic processes[1,2]. In the context of acute stress resulting from genotoxic chemotherapy, the initial adaptation is essential for tumor cell survival and may allow for subsequent development of chemoresistance through additional genetic and/or non-genetic mechanisms.

An integrator of extracellular signals and cellular response, the mTOR pathway is critical for cellular adaptation to environmental changes. Inhibition of mTOR induces diapause, a reversible arrest in the development of mammalian embryos in response to unfavorable environments[3]. Analogous to its role in regulating physiological diapause, mTOR inhibition may induce a reversibly paused state in cancer cells, leading to drug tolerance and chemoresistance. mTOR inhibition has been shown to be chemoprotective in leukemia cells[4,5]. Multiple transcriptomic studies have also reported an association between mTOR inhibition and resistance to chemo- and immunotherapies, with post-treatment residual tumors showing decreased mTOR activity[6–9]. However, the causal and mechanistic role of mTOR inhibition in chemoresistance has yet to be systematically investigated.

Here, we have identified the mTOR pathway as a major determinant of chemosensitivity through a genome-wide CRISPR knockout

[1]Center for Immunotherapy Research, Houston Methodist Research Institute, Houston, TX 77030, USA. [2]Department of Medicine, Weill Cornell Medical College, New York, NY 10065, USA. [3]UCLA-Caltech Medical Scientist Training Program, David Geffen School of Medicine, University of California, Los Angeles, Los Angeles, CA 90095, USA. ✉e-mail: yli@houstonmethodist.org

screen. We show that mTOR inhibition causally leads to the emergence of a reversible drug-tolerant persister population with a senescence phenotype, which is dependent on autophagy and G2/M cell cycle arrest for survival.

## Results

### CRISPR screen identifies mTOR pathway as a regulator of chemosensitivity

To systematically identify the genetic determinants of chemosensitivity, we performed a genome-wide CRISPR knockout screen in a murine pancreatic cancer cell line using two genotoxic chemotherapeutic agents, gemcitabine and selinexor (Fig. 1a). Gemcitabine is a deoxycytidine analog that inhibits DNA synthesis[10], and selinexor is an inhibitor of chromosome region maintenance 1/exportin 1 (CRM1/XPO1), required for chromosome segregation and mitotic progression[11,12]. As wild type p53 may interfere with Cas9 function and confound interpretation of chemosensitivity[13,14], we utilized the 4292 cell line, derived from a mouse pancreatic cancer model induced by Kras$^{G12D}$ and p53$^{R172H}$ mutants[15,16]. 4292 cells expressing Cas9 (4292-Cas9) were transduced with the Brie sgRNA library containing 78,637 sgRNAs targeting 19,674 genes[17], selected with puromycin, and subsequently treated with the two chemotherapeutic agents for 12 days. The screens were performed using relatively low concentrations of the drugs (20 nM for gemcitabine, and 0.33 μM for selinexor), in order to identify in the same screen enriched and depleted genes corresponding to negative and positive regulators of chemosensitivity.

Following deep sequencing of sgRNA libraries, significantly enriched or depleted genes in each screen were identified based on the β score[18]. Notably, a large fraction of the identified genes was shared by the two screens (Fig. 1b, c; Supplementary Data 1a–c). Gene Ontology

(GO) term enrichment analysis was performed using EnrichR[19] on genes that were significantly changed in both screens. Enriched genes were mainly clustered into protein translation, RNA metabolism and macromolecule biosynthesis, while depleted genes were involved in the negative regulation of mTOR signaling and autophagy (Supplementary Data 1d, e). We specifically examined the β scores of genes involved in mTOR signaling (GO: 0031929), including structural components as well as positive and negative regulators of mTOR complexes. Among the 118 mTOR signaling genes included in the sgRNA library, the top fifteen enriched genes based on average β score of the two CRISPR screens were exclusively involved in the activation of the mTOR pathway, including *Mtor*, *Mios*, *Telo2*, and *Raptor*; while top ten depleted genes were negative regulators of mTOR signaling, including *Tsc1*, *Tsc2*, *Stk11*, and *Nprl2/3* (Fig. 1d). Importantly, most enriched/depleted genes were related to the mTORC1, but not the mTORC2 complex (Supplementary Fig. 1a). To further validate our findings, we reanalyzed a published CRISPR screen performed in REH leukemia cells carrying a *TP53* mutation[8], and confirmed that reduced mTOR pathway activity indeed confers resistance to various DNA damaging agents (Supplementary Fig. 1b). Hence, our CRISPR screen identifies mTOR pathway as a prominent determinant of chemosensitivity.

### mTOR inhibition promotes the emergence of drug-tolerant persisters

To examine the role of mTOR in regulating chemosensitivity, we compared the efficacy of single-agent chemotherapy to chemotherapy plus mTOR inhibitors (mTORi). 4292 cells were treated with single-agent gemcitabine or selinexor to determine the concentrations that would completely eradicate tumor cells within seven days. We then treated the cells with each drug as a single agent at the killing con-

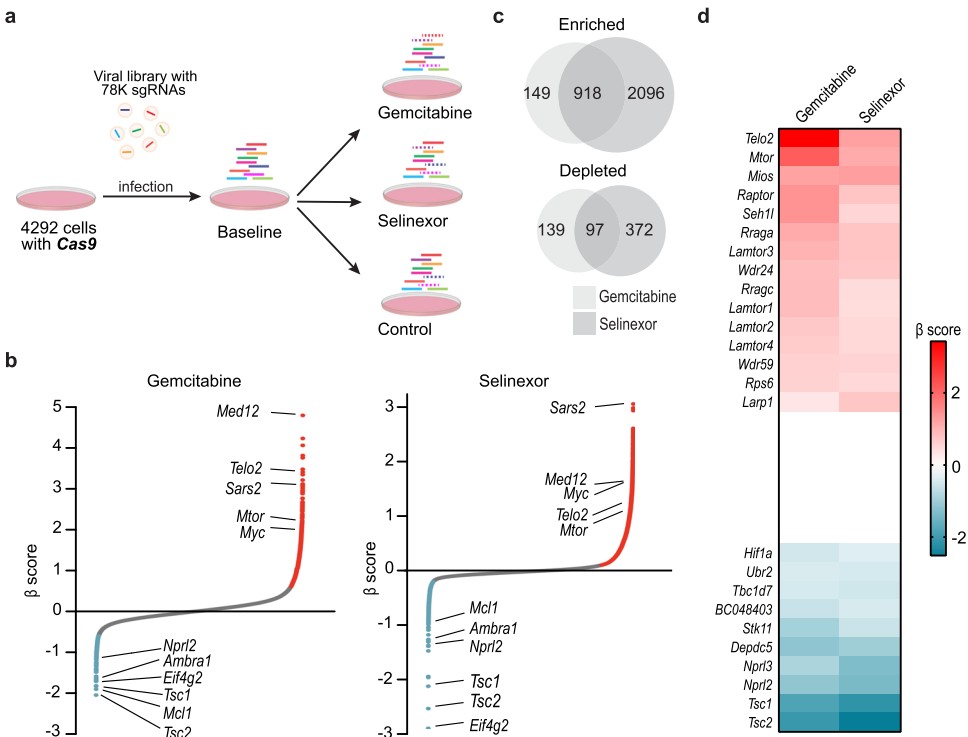

**Fig. 1 | Genome-wide CRISPR screening identifies the mTOR pathway as a determinant of chemosensitivity. a** CRISPR knockout screens in 4292 cells. Dashed bars indicate depleted sgRNAs. **b** Gene ranking based on β scores in the two screens. Representative top enriched or depleted genes were labeled on the plots. Red and blue indicate significant gene enrichment or depletion with *p* < 0.05. Two-sided permutation test. False discovery rate (FDR) correction of the p-value was

performed by the Benjamini-Hochberg procedure. **c** Overlap of significantly enriched or depleted genes in gemcitabine and selinexor screens. **d** Enrichment of positive, and depletion of negative regulators of mTOR signaling in gemcitabine and selinexor screens. The mTOR pathway genes were ranked based on the average β score in the two screens. Shown are the top 15 enriched and 10 depleted genes.

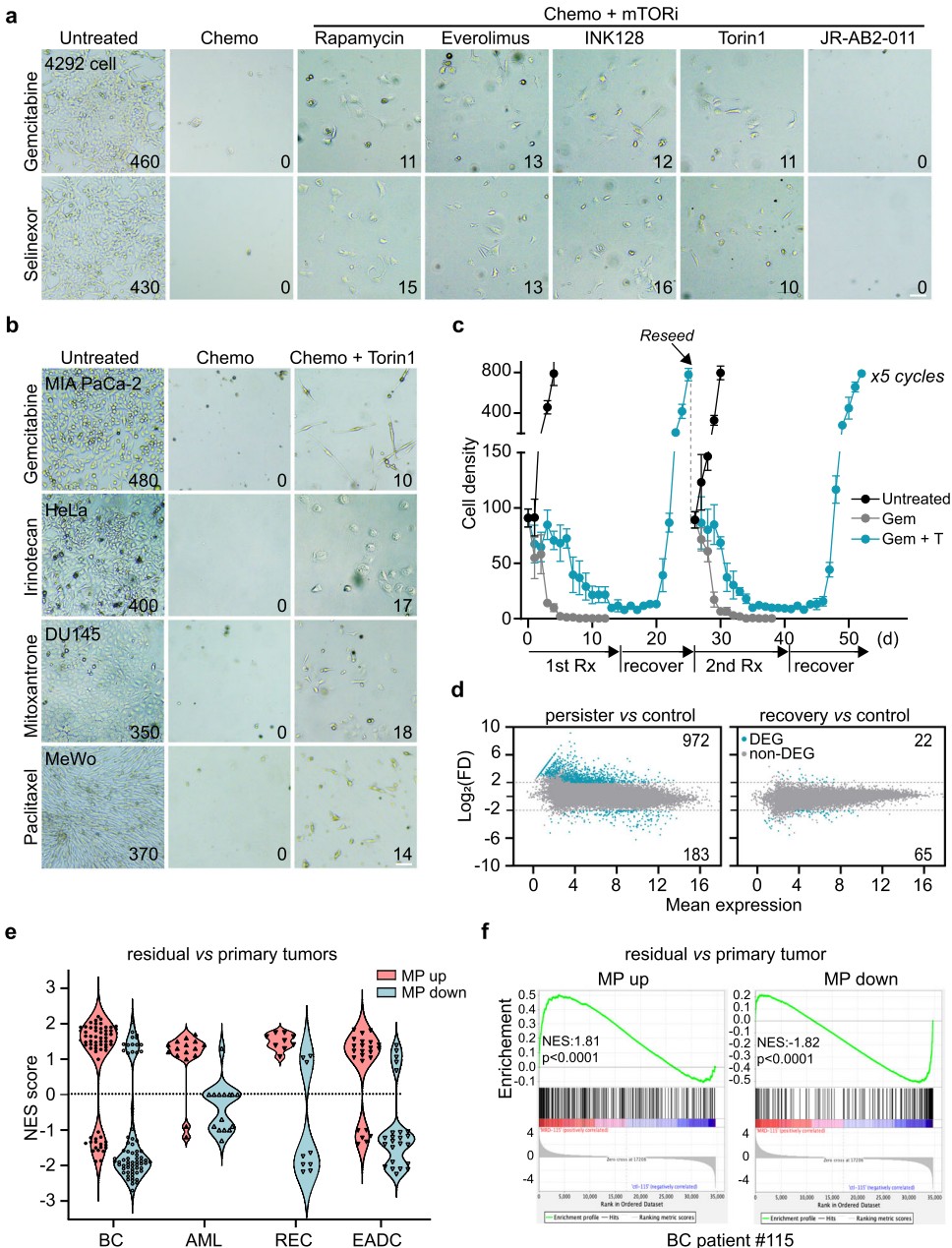

**Fig. 2 | mTOR inhibition induces reversible chemoresistant persisters. a** 4292 cells were treated with chemotherapeutic agents (2 μM gemcitabine, 2 μM selinexor), or chemotherapeutic agents plus a panel of mTOR inhibitors (rapamycin, everolimus, INK128, and Torin1 at 100 nM, and JR-AB2-011 at 1 μM) for seven days. Shown are representative images of cell survival and morphology from three biological replicates. The average cell count per image is indicated in the lower right corner. Scale bar, 100 μm. **b** Microscopic images of human cancer cells treated with Torin1 plus chemotherapeutic agents (100 nM gemcitabine for MIA PaCa-2 and 10 nM paclitaxel for MeWo for nine days; 60 μM irinotecan for HeLa and 100 nM mitoxantrone for DU145 for seven days). The images are representative of three biological replicates. The average cell count per image is indicated in the lower right corner. Scale bar, 100 μm. **c** Reversibility of chemoresistance in persister cells. Treatment (Rx), recovery, and reseeding were repeated for five cycles. Y axis represents cell number in a 50X field (0.64 mm² of culture area). For each data point, a minimum of three independent fields were counted and cell numbers were plotted as mean ± SEM. **d** Transcriptomic comparison of persister *vs* control and recovery *vs* control. Blue dots represent differentially expressed genes (DEG) that change at least 4-fold, while grey dots indicate non-DEG. Two-sided Wald test. *p* < 0.001, adjusted by the Benjamini–Hochberg procedure. **e** Violin plots of the GSEA results in residual *vs* primary tumors for multiple human cancer types using the MP signature. BC: breast cancer (GSE87455)[21], AML: acute myeloid leukemia (GSE40442)[24], REC: rectal cancer (GSE15781)[23], EADC: esophageal cancer (GSE165252)[22]. NES: normalized enrichment scores. **f** Representative GSEA plot for patient #115 in the breast cancer dataset (GSE87455)[21]. Kolmogorov–Smirnov test, *p* < 0.0001. Source data are provided as a Source Data file.

centrations, as well as in combination with rapamycin, and monitored their survival over the course of two weeks. Interestingly, combined treatment, while killing a large fraction of the cells, led to a subpopulation of survivors (Fig. 2a). Emergence of the drug-tolerant population was further confirmed using additional rapalogs specific to mTORC1 (temsirolimus and everolimus), as well as mTORC1/2 dual

inhibitors (Torin1 and INK128). In contrast, the mTORC2-specific inhibitor, JR-AB2-011, failed to induce the drug-tolerant subpopulation, indicating that the reduced mTORC1, but not mTORC2 activity, leads to chemoresistance (Fig. 2a). The function of these mTORi were confirmed by Western blot analysis (Supplementary Fig. 2a-d). In the presence of drug treatment, surviving tumor cells remained dormant

and viable for over three weeks (the longest time of observation), herein referred to as "persisters".

To determine whether the induction of persisters following mTOR inhibition is a general phenomenon, we treated human pancreatic cancer MIA PaCa-2 cells with a diverse panel of chemotherapeutic agents. mTOR inhibition led to the emergence of persisters in the majority of agents tested, including doxorubicin (anthracycline), selinexor (XPO1 inhibitor), paclitaxel (antimicrotubule), etoposide (topoisomerase II inhibitor), mitoxantrone (anthracycline), and irinotecan (topoisomerase I inhibitor) (Supplementary Fig. 2e). The effects of mTOR inhibition were further evaluated in an additional panel of 29 human cancer cell lines from diverse tissue origins, including pancreatic, breast, prostate, colon, liver, lung, ovarian, cervical cancers and melanoma. Treatment with chemotherapeutic agents and mTOR inhibitors led to the emergence of persisters in the majority of these cells. Cell lines that failed to show the drug-tolerant subpopulation were mostly TP53 wild type (MCF7, HCT116, H460, A549, A375, and HepG2), indicating that induction of the drug-tolerant state by mTOR inhibition may be a common feature of human cancers with TP53 mutations (Fig. 2b, Supplementary Table 1, Supplementary Data 2). To further examine the role of TP53 status in the persister phenotype, we lentivirally expressed p53[R273H] and p53[R249S] mutants in A375 and HCT116 cells[20]. Interestingly, treatments with gemcitabine or paclitaxel plus Torin1 in the two cell lines transduced with mutant p53 led to the emergence of persisters (Supplementary Fig. 2f, g).

We focused on the MIA PaCa-2 model to further examine the drug tolerant phenotype. Following drug removal, a fraction of the persisters resumed proliferation (Supplementary Fig. 3a). Recovered cells were sensitive to gemcitabine and completely eradicated by gemcitabine treatment alone, while combined treatment with gemcitabine and Torin1 again led to the emergence of survivors. The treatment-recovery was continued for five cycles and persisters consistently appeared following each cycle of chemotherapy plus mTOR inhibition (Fig. 2c). The persister phenotype was further validated in three independent single-cell clones derived from MIA PaCa-2 cells, and induction of persisters and phenotypic reversibility was similarly observed in each clone (Supplementary Fig. 3b). These results indicate that the drug-tolerant cells arise de novo upon non-genetic adaptation and are unlikely to derive from rare clones with preexisting genetic mutations. Additionally, we compared the transcriptomes of control, Torin1-treated, persisters, and recovered cells (persisters that were cultured in drug-free media for an additional 11 days) (Supplementary Data 3a–d). Interestingly, the transcriptional profiles of Torin1-treated cells and persisters are highly depleted for MYC target gene expression, G2/M cell cycle, mitotic transition, and mTOR signaling, according to the Gene Set Enrichment Analysis (GSEA) of the 50 hallmark gene sets (Supplementary Fig. 3c). While control and persisters differ drastically in their gene expression pattern, only modest differences are observed between the control and recovered cells (Fig. 2d). These results suggest that persisters largely revert to the cellular state of the pre-treatment control following drug removal. The acquired resistance following mTOR inhibition is, therefore, unlikely mediated by stable genetic changes, but a consequence of an mTOR inhibition-induced adaptive response to chemotherapeutic stress.

To examine the relevance of the persister phenotype in clinical cancer treatments, we derived an mTOR-regulated persister (MP) signature by identifying the common transcriptomic changes in the Torin1-treated vs control and persister vs control comparisons. The MP signature consisting of 403 upregulated (MP up) and 213 down-regulated genes (MP down) was used in GSEA of public transcriptomes of residual tumors following chemotherapy from several cancer types, including breast, rectal, and esophageal cancers[21–23], and acute myeloid leukemia[24]. Significantly, the upregulated MP genes were enriched, while the downregulated MP genes were depleted in the

transcriptomes of residual tumors (Fig. 2e, f; Supplementary Data 4a, b), suggesting that the mTOR-regulated persister state may be responsible for the minimal residual disease phenotype following clinical cancer chemotherapy.

## mTOR Activation increases chemosensitivity

Having demonstrated that mTOR inhibition promotes the development of drug-tolerant persisters, we further examined whether activation of mTOR pathway could conversely increase chemosensitivity. We used CRISPR to knock out two genetic suppressors of mTOR, TSC1, and TSC2, in MIA PaCa-2 cells. Western analysis confirmed the loss of TSC1 or TSC2 expression and activation of the mTORC1 pathway as indicated by increased S6 phosphorylation (Fig. 3a). We labeled the wild type clones with red fluorescent protein (RFP) and TSC1/TSC2 knockout clones with green fluorescent protein (GFP), and mixed at a one to one ratio in a multicolor competition assay (MCA). Treatment with gemcitabine or selinexor selectively eradicated the knockout clones as shown by the disappearance of GFP positive cells, while the GFP/RFP ratio did not change significantly in the untreated populations (Fig. 3b, c).

To assess the chemosensitivity in vivo, we transplanted wild-type and TSC2[−/−] MIA PaCa-2 cells into immunodeficient NOD scid gamma (NSG) mice. Despite four weeks of gemcitabine treatment, wild-type tumors expanded four-fold according to bioluminescence imaging (BLI) quantification. Interestingly, significant shrinkage was observed in the majority of TSC2[−/−] tumors (Fig. 3d, Supplementary Fig. 4a, Supplementary Data 5). The increased chemosensitivity upon mTOR activation translated into better survival of mice bearing TSC2[−/−] tumors following chemotherapy compared to those with wild-type tumors, whereas the non-treated cohorts did not show the difference in survival (Fig. 3e). Furthermore, analysis of the reverse phase protein array (RPPA) data in The Cancer Genome Atlas (TCGA) project[25,26] points to a significant survival benefit in pancreatic cancer patients with high mTOR protein expression in their tumors (hazard ratio = 0.381, p = 0.022). In particular, high levels of phospho-mTOR-S2448, indicative of mTOR kinase activity, is remarkably correlated with better overall survival (hazard ratio = 0.29, p = 0.0014) and progress-free survival (hazard ratio = 0.414, p = 0.00049) in pancreatic cancer, as well as multiple other cancer types, including lung, liver, kidney, prostate, cervical cancers, and melanoma (Fig. 3f, Supplementary Fig. 4b). Taken together, mTOR inhibition promotes drug tolerance, while activation of the mTOR pathway increases chemosensitivity, and is predictive of improved clinical survival in cancer patients.

## Persister cells display a senescence phenotype

Persisters have a distinctly enlarged and flattened morphology, reminiscent of the human fibroblasts undergoing replicative senescence[27]. We, therefore, examined these cells for the molecular changes and phenotypic features associated with senescence. Flow cytometric analysis detected an approximately 4- and 6-fold increase in the forward scatter (FSC) and side scatter (SSC) in the persisters compared to the control, consistent with the characteristic cell size and granularity of senescent cells (Supplementary Fig. 5a). Torin1-treated cells demonstrated a modest increase in FSC and SSC compared to the non-treated control. Canonical senescence-associated β-galactosidase (SABG) assay revealed extensive staining in the persisters but not in control or Torin1-treated cells (Fig. 4a, b, Supplementary Fig. 5b, c). Flow cytometric quantification of SABG activity with the 5-dodecanoylaminofluorescein di-β-D-galactopyranoside (C12-FDG) substrate[28] showed a 4-fold higher mean fluorescence intensity (MFI) in persister cells, while Torin1-treated cells did not differ from the control (Fig. 4c, d, Supplementary Fig. 5d). The senescent phenotype was also confirmed by drastic induction of p21 and γH2AX and silencing of lamin B1 protein expression in persisters as shown by Western analysis (Fig. 4e). These markers remained unchanged in Torin1-

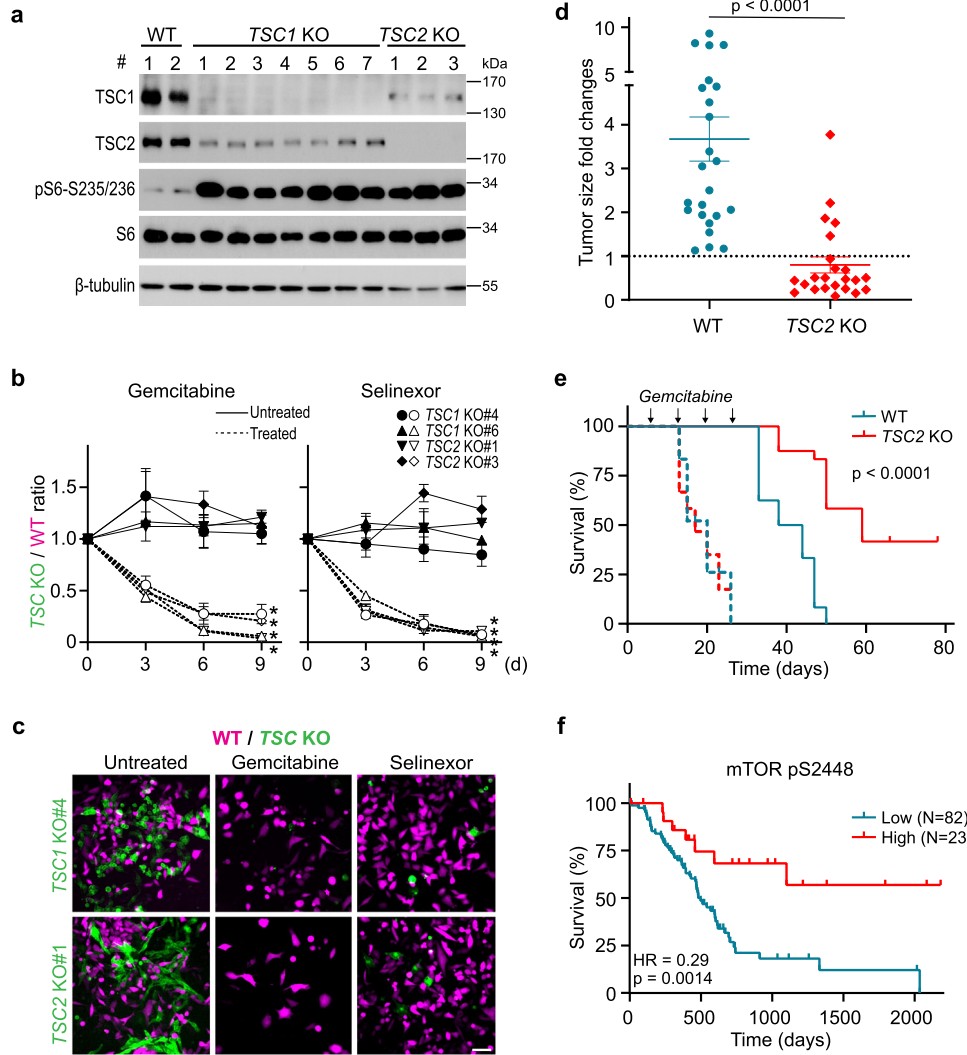

**Fig. 3 | mTOR activation confers chemosensitivity in vitro and in vivo. a** Western analysis of single-cell clones of *TSC1* and *TSC2* knockouts derived from MIA PaCa-2 cells. Two wild-type (WT), seven *TSC1* knockout (*TSC1* KO), and three *TSC2* knockout (*TSC2* KO) clones were generated. TSC1, TSC2, phospho-S6-S235/236, and S6 proteins were probed to show genomic ablation and mTORC1 activity. β-tubulin serves as the loading control. Experiment performed with three biological replicates. **b** MCA of wild type mixed with *TSC1* or *TSC2* knockout clones in the presence or absence of 30 nM gemcitabine or 300 nM selinexor for three, six, and nine days. Data plotted as mean ± SEM. Experiments were performed with three biological replicates. Two-tailed Student's *T* test, *$p < 0.05$. The *p* values for untreated *vs* gemcitabine-treated *TSC1* KO#4, *TSC1* KO#6, *TSC2* KO #1, and *TSC2* KO#3 are 0.021, 0.018, 0.021, and 0.012. The corresponding *p* values for selinexor treatment are 0.043, 0.024, 0.016, and 0.018. **c** Representative images from three treated cells, indicating the lack of senescence induction upon mTOR

MCA biological replicates on day 6. Scale bar, 50 μm. **d** Fold changes in tumor size based on BLI quantification following gemcitabine treatment in wild type (*N* = 24) and *TSC2* knockout (*N* = 23) tumors grown in NSG hosts. Data plotted as mean ± SEM. Two-tailed Student's *T* test, *p* = 3.67e-06. **e** Survival of NSG mice bearing wild type and *TSC2* knockout tumors with (solid line) or without (dashed line) gemcitabine treatment for four weeks. *N* = 24 for wild type and *N* = 23 for *TSC2* knockout in the treatment group, and *N* = 10 for both wild type and *TSC2* knockout in the non-treated group. Log-rank (Mantel-Cox) test, *p* = 5.17e-08. **f** Survival analysis of human pancreatic cancer patients based on RPPA analysis of phospho-mTOR-S2448 in tumors. RPPA data from TCGA study were analyzed using the TRGAted application. *N* = 23 for the high group and *N* = 82 for the low group. Log-rank (Mantel-Cox) test, *p* = 0.0014. Source data are provided as a Source Data file.

inhibition. Additionally, persister cells displayed a typical senescence-associated secretory phenotype (SASP) as revealed by Luminex detection of secreted cytokines. A minimum of 4-fold induction by the combination treatments was observed in multiple cytokines commonly associated with SASP, including CCL2, CCL26, G-CSF, MIF, FGF2, and CXCL5 (Fig. 4f)[29–32]. Finally, analysis of our transcriptomic data set utilizing a highly specific senescence classifier[6], clearly categorized the persisters as the senescent population (score of 0.99), while control, Torin1-treated, and recovered cells did not show any sign of senescence (scores less than 0.01) (Fig. 4g). Taken together, drug-tolerant persister cells display a prominent senescence phenotype according to multiple independent assays. Importantly, persister cells recovered

and re-established the tumor cell population following drug removal with complete reversibility, demonstrating a cellular state distinctly different from the replicative senescence with irreversible cell cycle arrest[33,34].

## A small-molecule chemical library screen reveals survival mechanisms of persisters

To identify the survival mechanisms in persisters, we performed a chemical library screen in the persister model of MIA PaCa-2 cells treated with gemcitabine and Torin1 for nine days, using a small-molecule library with approximately 200 inhibitors for various druggable targets and cellular processes (Supplementary Data 6a). Each inhibitor was tested in a serial dilution in both parental and persister

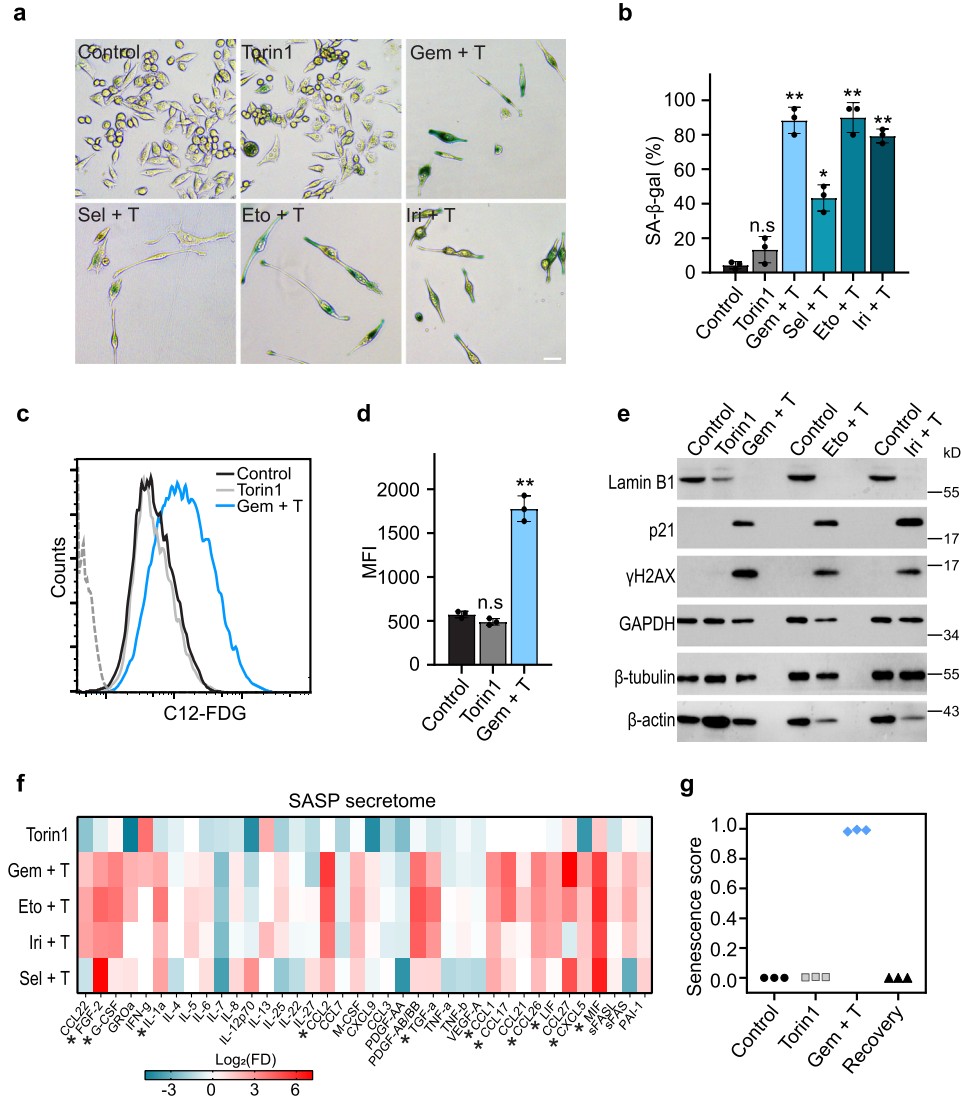

**Fig. 4 | Persister cells display a senescence phenotype. a** Representative images of SABG staining of MIA PaCa-2 cells treated for eight days with 100 nM Torin1, or Torin1 plus chemotherapeutic agents (100 nM gemcitabine, 2 μM selinexor, 6 μM etoposide, and 6 μM irinotecan). Scale bar, 50 μm. Experiments were performed with three biological replicates. **b** Quantification of SABG positive cells in control, Torin1, or Torin1 plus chemotherapy samples. The experiment was performed with three biological replicates. Data were plotted as mean ± SEM. Two-tailed Student's $T$ test. **$p < 0.001$; *$p < 0.01$; n.s: not significant. $p$ values for the Gem + T, Sel + T, Eto + T, and Iri + T treatments *vs* control sets are 5.16e-05, 0.001, 7.61e-05, and 8.92e-06. **c, d** Flow cytometric quantification of SABG activity using the C12-FDG fluorescent substrate in MIA PaCa-2 cells treated with 100 nM Torin1 or 100 nM gemcitabine plus Torin1 for nine days. Grey dashed line indicates the unstained control. Experiment was performed with three biological replicates. Data plotted as mean ± SEM. Two-tailed Student's $T$ test. **$p = 0.0002$; n.s: not significant. MFI: mean fluorescence intensity. **e** Western analysis of senescence markers in cells treated for nine days with Torin1 or Torin1 plus chemotherapeutic agents (100 nM gemcitabine, 6 μM etoposide, and 6 μM irinotecan). Experiment was repeated once with similar results. **f** Heatmap of cytokine secretion in MIA PaCa-2 cells following a nine-day treatment with 100 nM Torin1, or Torin1 plus chemotherapeutic agents (100 nM gemcitabine, 2 μM selinexor, 6 μM etoposide, and 6 μM irinotecan). Cytokines were detected using the Luminex assays. Asterisks indicate SASP (senescence-associated secretory phenotype) cytokines upregulated in persisters. **g** Senescence scores for control, Torin1-treated, persisters, and recovered cells analyzed using the Cancer SENESCopedia senescence classifier. RNA-seq experiments for each condition inlude three biological replicates. Source data are provided as a Source Data file.

cells. While most inhibitors did not affect persister survival, a few compounds modulating G2/M checkpoint, autophagy, and cell death pathways (apoptosis and ferroptosis[35]) selectively eradicated persisters, with minimal effect on the parental cells (Fig. 5, Supplementary Fig. 6a, b; Supplementary Data 6b). Inhibitors of autophagy, such as chloroquine, bafilomycin A1, SAR405 (VPS34 inhibitor), and MRT68921 (ULK1/2 inhibitor), efficiently eradicated persisters. Inhibition of multiple key G2/M cell cycle checkpoint regulators, including CHK1, ATR, and WEE1, also eliminated the persisters. Interestingly, multiple chemicals targeting CDK4/6, PDGFRs, PDK1, RSKs, and S6Ks, increased the persister population (Fig. 5, Supplementary Fig. 6a, b; Supplementary Data 6b). Kinases, such as PDK1, RSKs, and S6Ks, functionally converge

on the S6K/S6 downstream of mTORC1 pathway, suggesting that the emergence of persisters following mTOR inhibition may be due to reduced S6K function. Thus, our chemical screen identifies the multifaceted survival mechanisms in persisters.

Sensitivity to autophagy and G2/M checkpoint inhibitors was confirmed in MIA PaCa-2 persisters induced by treatment with selinexor, etoposide, irinotecan, and doxorubicin plus Torin1, with the addition of chemical inhibitors at the beginning of the treatment or following the emergence of persisters, equally blocked their survival (Supplementary Fig. 6c, d). Sensitivity to autophagy and G2/M checkpoint inhibitors was further demonstrated in additional cancer types (Supplementary Fig. 6e). In parallel to the sensitivity to

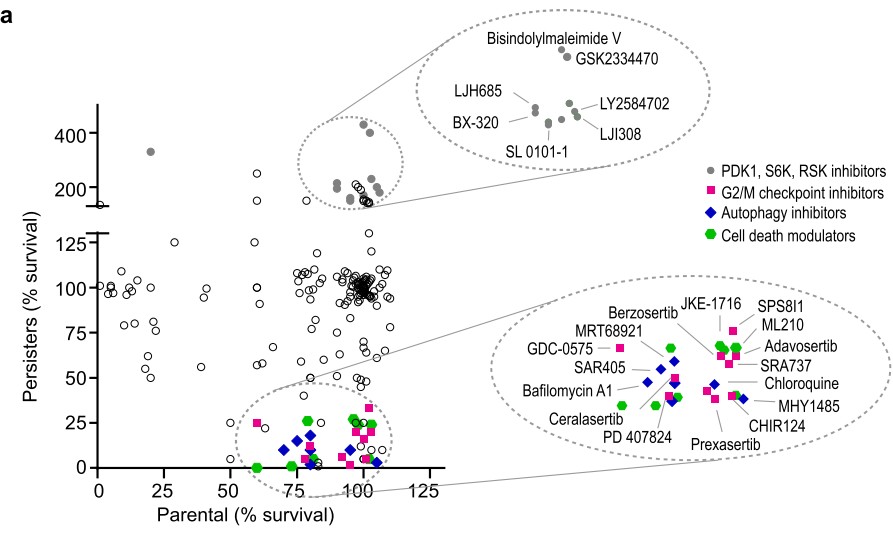

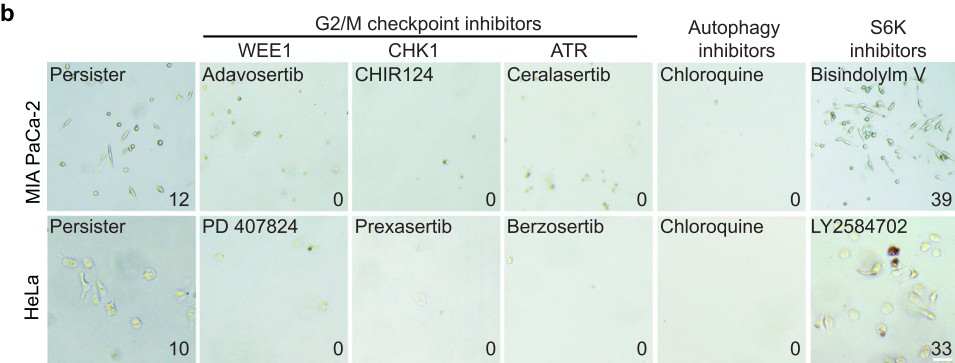

**Fig. 5 | Multifaceted survival mechanisms in persisters revealed by a small-molecule chemical screen. a** Effects of inhibitors on survival of MIA PaCa-2 persister and parental cells. Each chemical was plotted based on its effects on parental and persister survival. For each drug, survival data were normalized to untreated parental and persister cells. Dashed circles indicate four major groups of drugs modulating persister survival. **b** Representative images of persister survival following treatment with inhibitors in the chemical library. MIA PaCa-2 persisters were induced by the combined treatment with 100 nM gemcitabine and 100 nM Torin1 for nine days. HeLa persisters were induced by treatment with 60 µM irinotecan and 100 nM Torin1 for seven days. Experiments were performed in three biological replicates. Shown are inhibitors of G2/M checkpoint, autophagy, and S6K, including 100 nM adavosertib, 100 nM CHIR124, 10 µM ceralasertib, 10 µM chloroquine, and 30 µM bisindolylmaleimide V for MIA PaCa-2, and 3 µM PD 407824, 1 µM prexasertib HCl, 300 nM berzosertib, 10 µM chloroquine, and 1 µM LY2584702 for HeLa cells. The average cell count per image is indicated in the lower right corner. Scale bar, 100 µm.

autophagy inhibitors, we indeed observed strong evidence of autophagy induction in persisters based on Western analysis of protein markers (phospho-CREB, phospho-ATG14, and LC3), detection of LC3 puncta using a dual-color DsRed-LC3-GFP fluorescent reporter[36], and GSEA of transcriptomes (Supplementary Fig. 7a–e).

### mTOR inhibition potentiates G2/M cell cycle arrest to promote persister survival

Our chemical screen identified multiple G2/M checkpoint inhibitors targeting CHK1, ATR, and WEE1, effectively eliminating the persisters, indicating that G2/M cell cycle arrest is essential for persister survival. In a recent study, acute myeloid leukemia cells treated with chemotherapy persisted through a senescence-like state, which was effectively targeted by ATR inhibitors[34]. We, therefore, assessed the cell cycle distribution of persisters to unravel the survival mechanisms involving G2/M regulation. Flow cytometric analysis of persisters induced by multiple drug combinations showed prominent G2/M cell cycle arrest and polyploidy (Supplementary Fig. 8). Following release from the combined treatments, persisters resumed normal proliferation and displayed cell cycle profiles comparable to that of control cells. Using a fluorescent ubiquitination-based cell cycle indicator (FUCCI) reporter, we showed that treatment with chemotherapy plus Torin1 resulted in G2/M cell cycle arrest, while Torin1-treated cells were

predominantly in G1/S phase (Fig. 6a, b). In addition, induction of phospho-CHK1 (S345) and phospho-WEE1 (S642) pointed to activation of G2/M checkpoint in persisters (Fig. 6c, d). In contrast to the G2/M checkpoint activation, expression of G2/M transition genes was drastically downregulated in persister cells. Western blot analysis showed significant reduction in PLK1 and cyclin B1 protein levels in persisters, similar to Torin1-treated cells (Fig. 6c, d). Furthermore, GSEA of the transcriptomic data revealed depletion of transcripts involved in G2/M transition in both Torin1-treated and persister cells (Fig. 6e, f). Thus, the significant downregulation of genes involved in G2/M transition by mTOR inhibition could delay premature mitotic entry, thereby avoiding cell death by mitotic catastrophe. Concordantly, in a previous study, mTOR inhibition potentiated cell cycle arrest after irradiation through the transcriptional downregulation of cyclin B1 and PLK1, leading to increased survival of cancer cells following DNA damage[37]. Thus, regulation of the G2/M cell cycle arrest by mTOR inhibition mechanistically contributes to the survival of persisters.

### Discussion

We have identified the mTOR pathway as a key regulator of chemosensitivity through CRISPR screening. Chemotherapeutic treatment in the presence of mTOR inhibition induces a drug-tolerant persister population with a senescence phenotype, the survival of which is

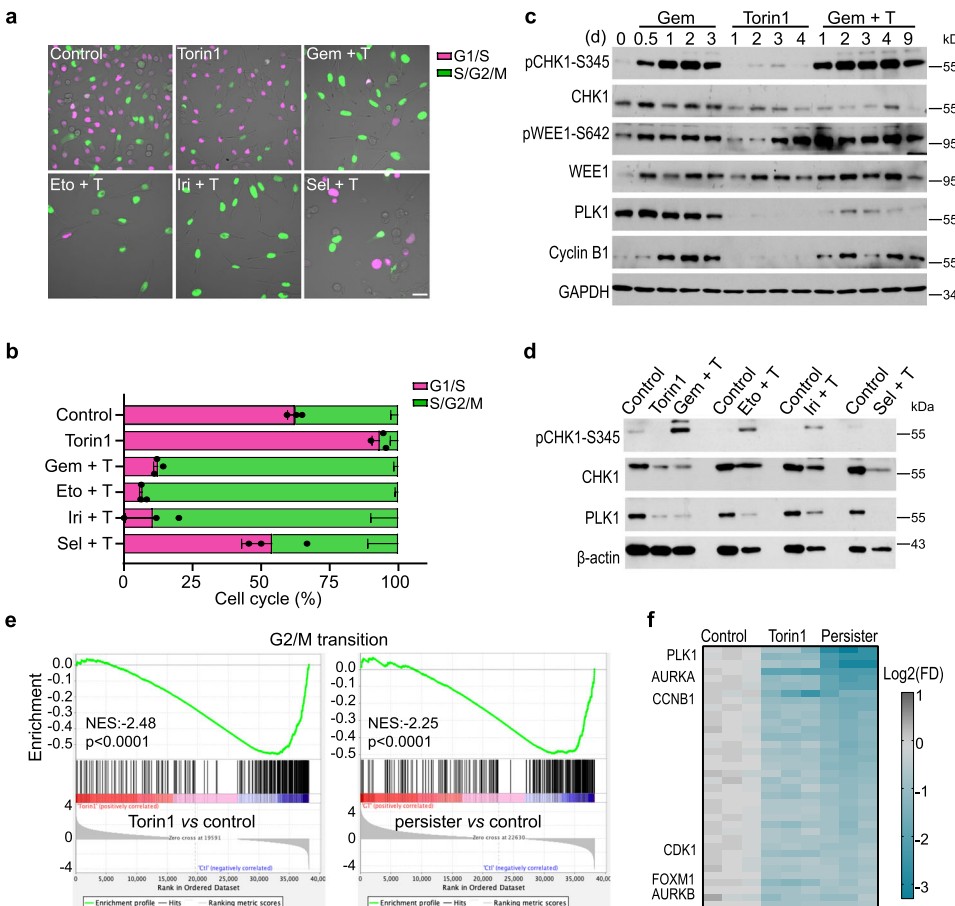

**Fig. 6 | mTOR inhibition potentiates G2/M cell cycle arrest and promotes persister survival. a**, **b** Cell cycle tracking with FUCCI reporter in control, Torin1-treated, and persister cells. Cells were treated with 100 nM Torin1, or Torin1 plus chemotherapeutic agents (100 nM gemcitabine, 2 μM selinexor, 6 μM etoposide, and 6 μM irinotecan) for nine days. Experiments were performed with three biological replicates. Data plotted as mean ± SEM. Scale bar, 50 μm. **c** Western blot analysis of key proteins involved in G2/M checkpoint activation and G2/M cell cycle progression. MIA PaCa-2 cells were treated with 100 nM gemcitabine, 100 nM Torin1, or the combination for the indicated days. The experiment was repeated once with similar results. **d** Western blot analysis of phospho-CHK1-S345 and PLK1 in MIA PaCa-2 cells treated with 100 nM Torin1 or Torin1 plus chemotherapeutic agents (100 nM gemcitabine, 2 μM selinexor, 6 μM etoposide, and 6 μM irinotecan) for nine days. The experiment was repeated once with similar results. **e** GSEA of G2/M transition signature (GOBP_CELL_CYCLE_G2_M_PHASE_TRANSITION from MSigDB) in Torin1-treated and persister cells. Kolmogorov–Smirnov test, $p < 0.0001$. **f** Expression of G2/M phase-specific transcripts in control, Torin1-treated, and persister cells. Genes were derived from the GSEA G2/M transition signature in Fig. 6e. Key regulators of G2/M transition are labeled. The color bar represents scale for log2 fold changes. Source data are provided as a Source Data file.

---

dependent on autophagy and G2/M cell cycle arrest. Conversely, activation of the mTOR pathway confers increased chemosensitivity in cancer cells with genetic ablation of *TSC1/2*. Additionally, activated mTOR signaling, as indicated by phospho-mTOR-S2448 levels in patients' tumors predicts better clinical survival among multiple cancer types. This notion is further supported by the reports of increased sensitivity to DNA damage-inducing therapies in cancer patients with mTOR activation caused by germline and/or somatic mutations of *TSC1/2*[38–41]. These observations demonstrate that mTOR activation confers a heightened chemosensitivity, despite its well-characterized oncogenic functions in tumorigenesis.

Persisters display a senescence phenotype similar to the physiological diapause, and distinct from replicative senescence with irreversible arrest[42]. mTOR is shown to causally regulate diapause during mouse blastocyst development[3]. More recently, colorectal cancer cells reportedly enter a diapause-like drug-tolerant persister state, concomitant with the reduction of mTOR signaling[7]. The functional roles of mTOR in physiological diapause upon nutritional deprivation and chemotherapy-induced senescence are likely similar, both involving induction of autophagy and cell cycle arrest[3]. Under physiological diapause, mTOR inhibition induces autophagy and dormancy to support long-term survival and pluripotency of blastocysts[3]. Under the stress of chemotherapy, mTOR inhibition promotes cancer cells to adopt a reversible drug-tolerant senescence state, which supports survival and allows for effective DNA repair, but incidentally, leading to chemoresistance and tumor recurrence.

The mTOR pathway is widely deemed as an appealing cancer therapeutic target[43–45]. However, the expectation has not been met in clinical trials. Among more than 500 trials using mTOR inhibitors as a single agent or in combination with other drugs, few have demonstrated clinical benefit[46–51]. Not surprisingly, FDA approval of mTOR inhibitors, including everolimus and temsirolimus, is only limited to the control of late-stage tumor expansion in a few specific cancers[51]. We observe that mTOR suppression promotes the induction of persisters in various cancer cells undergoing chemotherapy and dampens therapeutic efficacy. Similarly, others have shown that mTOR inhibition is chemoprotective in leukemia cells[4,5]. Thus, the use of mTOR inhibitors can be counterproductive in the context of chemotherapy. This may explain the general lack of efficacy of mTOR inhibitors in cancer clinical trials and argues against their widespread use in combination with DNA damage-inducing agents. In

this light, stimulation of mTOR signaling, but not its inhibition, may be the strategy of choice to reduce residual tumors and improve chemotherapeutic efficacy.

The paradoxical roles of oncogenes in regulating therapeutic responses is not limited to mTOR. Overexpression of *MYC* oncogene, while inducing aggressive tumor behavior, in fact leads to increased chemosensitivity[52–54]. Additionally, silencing *MYC* induces a diapause-like cellular state that is chemoresistant[55]. Indeed, *Myc* is one of the highly enriched hits in our CRISPR screen (Fig. 1b). Tumor suppressor activity may confer chemoresistance. Disruption of the p53 tumor suppressor enhances sensitivity to multiple chemotherapeutic agents, while its expression results in chemoresistance in various cancer types[56–59]. Similarly, attenuation of the RB tumor suppressor leads to aggressive tumorigenic growth accompanied by increased chemosensitivity[60–62]. Regulation of chemoresistance by oncogenes and tumor suppressors may mechanistically converge on senescence as MYC inhibition and p53/RB activation are known inducers of senescence[56,63–68]. Depending on the cellular contexts, suppression of oncogenes or activation of tumor suppressors may not result in tumor eradication via various forms of cell death but instead lead to tumor persistence through a reversible senescence state, thus promoting tumor recurrence. The seemingly paradoxical roles of oncogenes and tumor suppressors in regulating tumor growth and therapeutic sensitivity may be a common theme.

## Methods
### Mice
This study complies with all the relevant ethical regulations. All animal experiments were approved by the Institutional Animal Care and Use Committee (IACUC) at Houston Methodist Research Institute and were performed in accordance with institutional and national guidelines. Approximately four million firefly luciferase-labelled MIA PaCa-2 WT and *TSC2* KO cells were mixed with Matrigel and transplanted sub-cutaneously in 6–8 week old NSG female and male hosts ($n = 24$ for wild type and $n = 23$ for *TSC2* KO). Tumors were allowed to grow for three weeks prior to gemcitabine treatment (100 mg/kg, twice per week) for four weeks. Post-treatment survival was monitored for six weeks. BLI was performed on an IVIS Lumina III platform. Data were analyzed using the Living Image software (v4.2). In addition, 10 mice were included for wild type and *TSC2* KO tumors in the non-treated control cohorts. All mice were maintained under standard conditions, at ambient temperature, 60% humidity, 12 h light/dark cycles and received a standard diet and water *ad libitum*. Mice were euthanized by $CO_2$ exposure followed by cervical dislocation when the tumor diameter reached 1.5 cm, the maximal tumor size allowed by the IACUC committee.

### Cell lines and chemical reagents
The 4292 murine PDAC cell line was a gift from Dr. Marina Pasca di Magliano[15]. For human cell lines, MIA PaCa-2 (#85052806), PANC-1 (#87092802) and SCC47 (#SCC071) were from Sigma, MDA-MB-231 (#36) and SK-OV-3 (#43) were from MD Anderson, HeLa (#CCL-2), DU145 (#HTB-81), MeWo (#HTB-65), BxPC3 (#CRL-1687), Capan-2 (#HTB-80), LNCaP (#CRL-1740), C4-2B (#CRL-3315), 22Rv1 (#CRL2505), PC3 (#CRL-1435), T-47D (#HTB-133), MCF7 (#HTB-22), SK-BR-3 (#HTB-30), PLC/PRF/5 (#CRL-8024), SNU398 (#CRL-2233), Hep3B (#HB-8064), HepG2 (#HB-8065), A375 (CRL#1619), HCT116 (#CCL-247), SW480 (#CCL-228), H460 (#HTB-177), A549 (#CCL-185), H1299 (#CRL-5803), H441 (#HTB-174), U2OS (#HTB-96), AsPC-1 (#CRL1682) were from ATCC. Cell lines were authenticated by short tandem repeat profiling and tested as negative for mycoplasma contamination by PCR at the MD Anderson Cancer Center cell line core. With the exception of T47D, MDA-MB-231, C4-2B, DU145, H460, and 22Rv1, which were cultured in RPMI-1640, and PC3 in F-12K, all cells were cultured in DMEM supplemented with 10% fetal bovine serum and 1% penicillin/

streptomycin in a 5% $CO_2$ humidified incubator. Gemcitabine, 5-fluorouracil, paclitaxel, carboplatin, Torin1, rapamycin, everolimus, temsirolimus, bafilomycin A1, chloroquine, MHY1485 were purchased from Cayman Chemical. Selinexor, SAR405, and MRT68921 were obtained from Selleckchem. All chemicals for the small-molecule library screening were purchased from Cayman Chemical.

### CRISPR Cas9 genome-wide library screening
CRISPR library screen was performed in biological replicates following a published protocol[69]. 4292 cells were infected with lentiCas9-Blast (Addgene #52962). Following blasticidin selection, cells were infected at a low viral titer with the pooled mouse CRISPR lentiviral library containing 78,637 gRNAs targeting 19,674 genes (Addgene #73633-LV). Infected cells were selected with puromycin (1 µg/ml) for two days. The experimental pools were treated with gemcitabine (20 nM) or selinexor (0.33 µM) for 12 days. All cell pools were passaged every three days with at least 50 million cells per pool to ensure a minimum of 500x coverage. Following drug treatment, sgRNA libraries in the cell populations were isolated by PCR amplification and identified by Hiseq. Computational analysis of the sgRNA libraries was performed using MAGeCK (version 0.5.9.2) as reported[18].

### CRISPR knockout
The guide RNAs targeting specific genes were designed using CHOP-CHOP (https://chopchop.cbu.uib.no/). Two sgRNAs were cloned in pSpCas9(BB)−2A-GFP (PX458) (Addgene, #48138). MIA PaCa-2 cells were transiently transfected with an equimolar mixture of the sgRNAs using lipofectamine 3000 (Thermo Fisher). Three days after transfection, GFP positive cells were sorted for single cells on a BD FACS Aria cell sorter (BD Biosciences). Gene knockout was confirmed by genomic DNA PCR and Western analysis. The sgRNA sequences are:

hTSC1-ACGAGATAGACTTCCGCCACG
hTSC1-BAGTCGGTGGGAGACGACTAT
hTSC1-CGACGTCGTTGTCCTCACAAC
hTSC1-DTACCAATGATTCCACAGTCT
hTSC2-ACGTCTGCGACTACATGTACG
hTSC2-BAGGAGACGACTCGCTCGATG
hTSC2-CCGTCCGGACCGCGTCCTCTG
hTSC2-DCTGTCGCACCATCAACGTCA.

### Expression of p53 mutants
The two mutant constructs (pLenti6/V5-p53_R273H, Addgene #22934; pLenti6/V5-p53_R249S, Addgene #22935) were lentivirally delivered to the A375 and HCT116 cells[20]. Cells were selected with blasticidin (A375, 5 µg/mL; HCT116, 10 µg/mL) for one week before testing for the persister phenotype. Expression of the mutants was confirmed by Western blot analysis.

### SABG staining
MIA PaCa-2 and MDA-Panc-28 cells were plated in 24-well plates at 10,000 cells/well and treated with chemotherapeutic agents and Torin1 the next day. Following eight (MIA PaCa-2) or six (MDA-Panc-28) days of drug treatment, cells were stained using Senescence Cells Histochemical Staining Kit (Sigma #CS0030) according to the manufacturer's protocol. Nine images from each well were acquired and positively stained cells were manually counted.

### Luminex assay of secreted cytokines
Secretome analysis was performed using Luminex Human 80 plex (EMD-Millipore) at Stanford Human Immune Monitoring Center. Cell culture media were changed three days before sample collection. For each sample, levels of cytokines were normalized to the media volume and cell number. All cytokine detections were performed in technical duplicates.

## RNA-seq analysis

MIA PaCa-2 untreated cells (control), treated with 100 nM Torin1 (Torin1-treated), or 100 nM Torin1 and 100 nM gemcitabine (persisters) for nine days, and persister cells that were grown in drug free media for 11 days (recovery) were used for RNA-seq analysis. Total RNA was extracted using Qiagen RNeasy Plus kit (Qiagen). Library construction and total RNA sequencing were performed commercially (Novogene). For RNA-seq data analysis, kallisto (version 0.46.2)[70] was used to pseudoalign paired-end reads to a reference transcriptome with 100 bootstraps. The reference transcriptome used to build the kallisto index consisted of the "Protein-coding transcript sequences" and "Long non-coding RNA transcript sequences" FASTA files from Human Release 38 (GRCh38.p13) of GENCODE. The corresponding GENCODE human primary assembly GTF file was used as an annotation file. The R library, sleuth (version 0.30.0)[71], was used to produce normalized gene-level abundance estimates and perform differential gene expression analysis.

## GSEA using the mTOR-regulated persister signature and the Hallmark gene sets

To generate the mTOR-regulated persister signature from the RNA-seq data, we first derived the genes differentially expressed in Torin1-treated and persister cells (two-fold changes at an adjusted $p < 0.01$). The overlap of the two differentially expressed gene sets was defined as the mTOR-regulated persister signature, which includes both upregulated and downregulated genes (MP up and MP down) (Supplementary Data 4a). GSEA of the public datasets (GSE87455 for breast cancer[21], GSE165252 for esophageal cancer[22], GSE15781 for rectal cancer[23], and GSE40442 for acute myeloid leukemia[24]) was performed using our MP signature on post-chemotherapy residual tumors vs pretreatment primary tumors. The normalized enrichment scores (NES) for both MP up and MP down signatures in the three public data sets were plotted using GraphPad Prism (version 8.4.2). Comparison of the persister vs control and Torin1-treated vs control transcriptomes was carried out with the "Hallmark gene sets" (50 sets) from MSigDB collections. GSEA analysis was performed using GSEA_4.1.0 from http://www.gsea-msigdb.org/gsea/downloads.jsp

## Cell cycle analysis using FUCCI reporter

FUCCI reporter (Addgene #86849) was stably expressed in MIA PaCa-2 cells by lentiviral transduction. Cells were seeded in 24-well plates and treated the next day with indicated drugs for nine days. At least nine representative images of each condition were captured on an FV3000 confocal microscope equipped with Olympus FV31S software and analyzed using the Olympus CellSens Dimension package. Magenta and green colors indicate distribution in G1/S and S/G2/M phase, respectively.

## Survival analysis of patients' RPPA data in TCGA

Survival analysis of patients' samples was based on the reverse-phase protein array (RPPA) data on mTOR and phospho-mTOR-S2448 in The Cancer Genome Atlas (TCGA) project. Data analysis was performed using the TRGAted application with optimal cutoff[25].

## Western blot analysis

Cells were lysed on ice with CelLytic MT reagent (Sigma) supplemented with protease and phosphatase inhibitors. Primary antibodies: TSC1 (#6935), TSC2 (#4308), phospho-S6 (#4858), S6 (#2217), p21 (#2947), Lamin B1 (#12586), LC3 A/B (#12741), phospho-ATG14-S29 (#92340), ATG14 (#96752), phospho-CREB-S133 (#9198), CREB (#9197), phospho-CHK1-S345 (#2348), phospho-WEE1-S642 (#4910), WEE1(#13084), PLK1 (#4513), γH2A.X (#9718), phospho-S6K-S371 (#9208), S6K (#2708), phospho-4EBP1-S65 (#9451), 4EBP1 (#9644), phospho-AKT-S473 (#4060), AKT (#4691), p53 (#2527), phospho-CDC2-T15 (#9111), CDC2 (#9116), and phospho-ATR-S428 (#2853) from Cell signaling (all used at

a 1: 1000 dilution); β-tubulin (#10094-1-ap, 1: 4000), β-actin (#20536-1-ap, 1: 4000), and GAPDH (#10494-1-ap, 1: 10000) from ProteinTech; ATR (#A300-137A-T, 1: 1000) from Bethyl Lab, and CHK1 (#sc-8408, 1:1000) and Cyclin B1(#sc-245, 1: 1000) from Santa Cruz.

## Flow cytometry analysis

For cell cycle distribution, cells were resuspended in PBS, fixed with ice-cold ethanol, and treated with RNase and propidium iodide, and analyzed on a BD FACS LSRII flow cytometer (BD Bioscience). Quantification of SABG enzymatic activity followed a previously described protocol[72]. Briefly, cells were treated with bafilomycin A1 (100 nM) for one hour and stained with C12-FDG (33 μM) for two hours at 37°C in a 5% CO2 incubator before analysis. Data were analyzed using FlowJo software (v10.8.1) (Tree Star).

## Autophagy reporter assay

pQCXI Puro DsRed-LC3-GFP (Addgene #31182) reporter was introduced in MIA PaCa-2 cells by retroviral infection. Cells expressing the reporter were seeded in 4-well-chambered coverslip (ibidi #80426) and treated with the indicated drugs for nine days. Images were acquired using an FV3000 confocal microscope. For each condition, LC3 puncta in at least 50 cells were counted.

## Multicolor competition assay

MIA PaCa-2 wild type (WT) and TSC1/TSC2 knockout (KO) cells were transduced with retrovirus expressing RFP (pQCXIP-turboRFP, Addgene #73016) or EGFP (pQCXIP-EGFP-F, Addgene #73014). MIA PaCa-2 WT-RFP and TSC1/TSC2 KO-GFP cells were mixed 1:1. One day after seeding (d0) and 3, 6, and 9 days after chemotherapy, cells were imaged on a Leica DMi8 fluorescence microscope with a 10X objective using LAS X (v3.7.4) software. RFP and GFP positive cells were counted and the relative cell fitness was calculated from the ratio of RFP/GFP positive cells normalized to the d0 value.

## Small-molecule chemical library screen

Approximately 200 chemical inhibitors covering major cell signaling and survival pathways were dissolved in DMSO or PBS. Inhibitors were typically tested at a 1:3 dilution from 10 μM to 100 nM, although some inhibitors were tested at concentrations as low as 30 nM or as high as 100 μM. Parental cells (single-agent) and persisters (triple-agent) were treated with individual inhibitors for nine days, and their response at each concentration was recorded. Cell survival was monitored daily. Any concentrations leading to significant changes in viability of either parental cells or persisters were recorded and the lowest concentration were reported in Fig. 5 and Supplementary Data 6. If both parental cells and persisters were killed, the experiment would be repeated using lower concentrations. For inhibitors reported in other studies, we consulted the original publications for the typical concentration ranges and revised our design accordingly.

## Statistical analysis

Two-tailed Student's $t$ test was used to analyze the MCA data. Cell death rates among different treatment groups were analyzed using ANOVA with Tukey's test. Results were presented as mean ± SEM. Animal survival was compared by Kaplan-Meier analysis with the log-rank (Mantel-Cox) test. For the gene ranking from CRISPR screen, two-sided $p$-values were computed by a permutation test in the MAGeCK software, and FDR correction of the p-value was performed by the Benjamini-Hochberg (BH) procedure. For the comparison of transcriptome pattern and differential gene expression analysis, two-sided $p$-value were computed by the Wald test in the sleuth software and were adjusted by the BH procedure. One-sided Fisher's exact test was performed in Enrichr for the GO term enrichment analysis of enriched/depleted genes. Microsoft Excel 2015 and GraphPad Prism v8.4.2 were used for statistics.

## Reporting summary

Further information on research design is available in the Nature Portfolio Reporting Summary linked to this article.

## Data availability

Sequencing data generated in this study have been deposited in GEO with accession number GSE162065 for the CRISPR screens and GSE189764 for the RNA-seq experiment. The publicly available residual tumor data used in this study are available in the GEO database under accession codes: GSE87455[21], GSE165252[22], GSE15781[23], GSE40442[24]. The RPPA data in TCGA project used for survival analysis (TRGAted application) were from the TCPA portal developed by MD Anderson Cancer Center[25,26]. The remaining data are available within the article, supplementary information or source data file. All source data are included within the paper. Source data are provided with this paper.

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

## Acknowledgements

We thank Dr. Marina Pasca di Magliano for generously providing the 4292 murine pancreatic cancer cell line, Matthew Vasquez for assistance with confocal microscopy, Dr. Dawei Zou for help with flow cytometry, and Dr. Yael Rosenberg-Hasson for assistance with cytokine assays. This work was supported in part by NIH K22CA207598 (Y.Li.), CPRIT RP200472 (Y.Li.), and NIH T32GM008042 (D.K.S.; UCLA-Caltech Medical Scientist Training Program).

## Author contributions

Y.Liu performed in vitro studies and chemical screens, and analyzed the public data sets. N.G.A. carried out CRISPR screens and in vivo studies. D.K.S. analyzed the CRISPR screen and RNA-seq data. Y.Li conceived and supervised the project. Y.Liu, N.G.A., and Y. Li wrote the manuscript.

## Competing interests

The authors declare no competing interests.
