## [Peer Review File new · Nature Communications]

mTOR inhibition attenuates chemosensitivity through the induction of persistersReviewers' Comments:

Reviewer #1:

Remarks to the Author:

The manuscript by Liu et al. uses functional genomics to identify genes conferring resistance or sensitivity to chemotoxic agents in cancer cells. They screen a genome-wide CRISPR knockout library in p53/K-Ras mutant cells treated with chemotoxic agents. They reveal that the loss of function mutations in the mTOR pathway favors the persistence of a pool of senescent-like cells that can re-grow in favorable conditions. The data are interesting and well controlled. The novelty is somehow limited, as several studies already pointed that mTOR inhibition may be chemoprotective and radioprotective (including papers by A. Amon's lab not cited). More information would be required to better understand the precise cellular adaptations to these chemotoxic and genetic treatments.

- 1) After the genetic screen (Figure 1), are all the mutations homozygous or some are heterozygous? It would be surprising if the homozygous knockout of RpS6 and mTOR maintained viable cells.
- 2) Kinetics analysis at early time points should be provided after chemotoxic treatment alone (Gem and Sel, Figure 2). Are some treated cells undergoing senescence? Or exclusively apoptosis? Is Torin1 treatment shifting from apoptosis to senescence?
- 3) Similarly, kinetics analysis at early time points should be provided during the recovery phase. How is this reversible senescence characterized? Do all the persist cells recover? Or just a sub-population? Some kinds of single cell tracing should be assessed.
- 4) Is it possible that Cdk4 and Cdk6 inhibitors protect because the cells arrest in G1 phase and are not exposed to the replication stress of the chemotoxic agent? Or do the cells treated with Cdk4/6 inhibitors also accumulate in G2 phase? How is it possible that torin1 treatment arrests cells in G1 but the combined Gem + Torin treatment in G2? Does the same happen with S6K inhibitors? The S6K inhibitors should not affect cell cycle progression but rather cell size.

Reviewer #2:

Remarks to the Author:

Liu et al present data supporting the notion that mTORC1 inhibition leads to increased autophagy and a G2/M cell cycle arrest and that this facilitates a subpopulation of tumor cells to survive chemotherapy allowing subsequent regrowth.

Their observations are interesting and thought provoking. However, they also stand in the face of an enormous number of publications suggesting that mTORC1 inhibition in combination with other anti-cancer chemotherapeutics is a valid treatment strategy. The authors should discuss this disconnect in the Discussion.

General comments:

For figures containing microscopic images, scale bars should be defined in the figure legend. For figures containing color bars, it may be better to indicate whether it is fold change or log₂(fold change) on the figure rather than in the figure legend for ease. Figure legends were often missing crucial information, such as concentrations of all compounds used or time points of the experiments (all marked in the excel sheet).

They used quite a high number of chemotherapeutic agents or other small molecule inhibitors on a variety of cells. How did they decide the working concentration for all these compounds per cell type? Did they make preliminary experiments with variable concentrations and landed on these? If yes, how were they tested, what were the selection criteria? If no, do they have solid evidence from literature that these are the "standard" concentrations widely used in the field for each cell type?

Although they have acceptable reasons to use different concentrations of a certain compound in different experiments (like Crispr screen vs. other cell culture experiments), there are identical experiments having used different concentrations of the same compound, or more commonly, different time points! There is very low temporal consistency among persist-cell generating experiments with no given explanation.

They show no biochemical proof for the inhibitors they used (i.e. phospho-western blots etc.). Did they test the inhibition efficiency of said compounds at the given concentrations and time points? They do not mention the number of replicates for most of the experiments. How many times did

they replicate the "Persister cell generation +/- various compounds" experiments? They should include this information in EVERY figure legend.

Experimental detail is lacking – see Xlsx file

Specific comments (underlined text = experiments need to be performed):

Figure 2A: They show no validation of the inhibition efficiency for any of the compounds. They should show via phospho-immunoblots that 1) the compounds inhibit mTORC1 and/or mTORC2 in untreated cells at various concentrations spanning their final working concentration, and 2) mTORC1 (and also maybe mTORC2) activity is reduced in Chemo+mTORi samples compared to untreated or recovered cells. Likewise, they can show mTORC1/2 re-activation in recovering cells in a time course.

Figure 2D: Figure legend says "Black dots represent differentially ...". Those dots in the figure are blue.

Figure 3B: Having different colors for WT and TSC mutants would make it easier and quicker to understand. They can match the colors with Supp. Figure S4A.

Figure 3D: What were the initial sizes of the WT vs TSC2-KO tumors? Fold change size difference may be misleading, as it is processed data. It is necessary (or better scientific practice) to present all four data sets: WT and TSC2-KO sizes before and after Gemcitabine treatment. It would be also good, if they could provide representative images from BLI or MRI, showing tumor sizes before and after treatment.

Figure 3E: They MUST include survival curves of WT and TSC2-KO xenograft injected mice without treatment. How do we know the effect of Gemcitabine treatment? What if TSC2-KO tumor mice already have a better survival???

Figure 5A: It may be useful to add "% survival" to the axes titles to understand easily. They did not mention which cell line was used for this experiment in the figure legend.

Figure 5B: Figure legend has Adavosertib treatment performed to HeLa cells, however the figure does not contain this treatment. Instead there is Benzosertib treatment which is not mentioned in the figure legend. Like mTORC1/2 inhibitors, they should validate the inhibition capacity of their WEE1, CHK1 and ATR inhibitors at given concentrations via phospho-immunoblots.

Figure 6A,B: Do they have quantification data from the other treatments than Gem+T? Are there G2/M ratio differences among these chemotherapeutic drugs? Does it correlate with the flow cytometry analyses in Supp. Figure 8B, and the difference in polyploidy & G2 accumulation?

Figure 6D: Total CHK1 levels need to be shown.

Supp. Figure S2A: Figure and figure legend indicates Selinexor, but main text says Gemcitabine (line 108).

Supp. Figure S2B: This panel is not related to where it is mentioned in the text (line 117).

Actually, there is no proper explanation of this panel in the Results.

Supp. Figure S2C: How consistent were the experimental conditions among all these treatments? Same concentration and time points were used? What was the "yes" or "no" selection criteria, the lack of which led to "not determined" phenotype?

Supp. Figure S3B: The axes may be labeled on the figure (probably "NES"..?). The top graph shows pathway dysregulations between Torin1 treated and control cells, and it seems to have a similar dysregulation pattern compared to persister vs. control cells graph. First of all, what are Torin1 samples? The RNA-seq was performed with Control, Persister and Recovery cells. Did they have an additional "only Torin1 treated" cells next to Persisters? Then, why are the expression profile of Torin1 samples is almost the same as Persisters? The associated text in the Results (lines 128-134), does not clearly explain this top panel. Lower panel is clear.

Supp. Figure S4A: Having different colors for WT and TSC mutants would make it easier and quicker to understand. In contrast to their claim, 5-FU and Paclitaxel treated cells do not seem to have an increased sensitivity. Same goes for Gemcitabine. Only Selinexor seems potentially significant. They should have done appropriate statistical analyses to be able to conclude that. The figure legend is missing info: the error bars are showing SD or SEM? In any case, judging from the variation, only Selinexor looks like a potential "real" effect. Also, why so low concentrations of Gem and Sel? Why 4-5 days treatment, and not 9-days like they most often did?

Supp. Figure S4B: The "high" and "low" p-mTOR staining criteria are available on the TGCA database? Or the authors came up with a quantitative strategy to decide this? If yes, how?

Supp. Figure S6B: Adavosertib or Chloroquine were added either at the beginning or 5 days after the start. What is the final day of the experiment? How many days after Day 5?

Supp. Figure S7A: Autophagic influx is usually measured by LC3-II / LC3-I ratio. Although the

they replicate the "Persister cell generation +/- various compounds" experiments? They should include this information in EVERY figure legend.

Experimental detail is lacking – see Xlsx file

Specific comments (underlined text = experiments need to be performed):

Figure 2A: They show no validation of the inhibition efficiency for any of the compounds. They should show via phospho-immunoblots that 1) the compounds inhibit mTORC1 and/or mTORC2 in untreated cells at various concentrations spanning their final working concentration, and 2) mTORC1 (and also maybe mTORC2) activity is reduced in Chemo+mTORi samples compared to untreated or recovered cells. Likewise, they can show mTORC1/2 re-activation in recovering cells in a time course.

Figure 2D: Figure legend says "Black dots represent differentially ...". Those dots in the figure are blue.

Figure 3B: Having different colors for WT and TSC mutants would make it easier and quicker to understand. They can match the colors with Supp. Figure S4A.

Figure 3D: What were the initial sizes of the WT vs TSC2-KO tumors? Fold change size difference may be misleading, as it is processed data. It is necessary (or better scientific practice) to present all four data sets: WT and TSC2-KO sizes before and after Gemcitabine treatment. It would be also good, if they could provide representative images from BLI or MRI, showing tumor sizes before and after treatment.

Figure 3E: They MUST include survival curves of WT and TSC2-KO xenograft injected mice without treatment. How do we know the effect of Gemcitabine treatment? What if TSC2-KO tumor mice already have a better survival???

Figure 5A: It may be useful to add "% survival" to the axes titles to understand easily. They did not mention which cell line was used for this experiment in the figure legend.

Figure 5B: Figure legend has Adavosertib treatment performed to HeLa cells, however the figure does not contain this treatment. Instead there is Benzosertib treatment which is not mentioned in the figure legend. Like mTORC1/2 inhibitors, they should validate the inhibition capacity of their WEE1, CHK1 and ATR inhibitors at given concentrations via phospho-immunoblots.

Figure 6A,B: Do they have quantification data from the other treatments than Gem+T? Are there G2/M ratio differences among these chemotherapeutic drugs? Does it correlate with the flow cytometry analyses in Supp. Figure 8B, and the difference in polyploidy & G2 accumulation?

Figure 6D: Total CHK1 levels need to be shown.

Supp. Figure S2A: Figure and figure legend indicates Selinexor, but main text says Gemcitabine (line 108).

Supp. Figure S2B: This panel is not related to where it is mentioned in the text (line 117).

Actually, there is no proper explanation of this panel in the Results.

Supp. Figure S2C: How consistent were the experimental conditions among all these treatments? Same concentration and time points were used? What was the "yes" or "no" selection criteria, the lack of which led to "not determined" phenotype?

Supp. Figure S3B: The axes may be labeled on the figure (probably "NES"..?). The top graph shows pathway dysregulations between Torin1 treated and control cells, and it seems to have a similar dysregulation pattern compared to persister vs. control cells graph. First of all, what are Torin1 samples? The RNA-seq was performed with Control, Persister and Recovery cells. Did they have an additional "only Torin1 treated" cells next to Persisters? Then, why are the expression profile of Torin1 samples is almost the same as Persisters? The associated text in the Results (lines 128-134), does not clearly explain this top panel. Lower panel is clear.

Supp. Figure S4A: Having different colors for WT and TSC mutants would make it easier and quicker to understand. In contrast to their claim, 5-FU and Paclitaxel treated cells do not seem to have an increased sensitivity. Same goes for Gemcitabine. Only Selinexor seems potentially significant. They should have done appropriate statistical analyses to be able to conclude that. The figure legend is missing info: the error bars are showing SD or SEM? In any case, judging from the variation, only Selinexor looks like a potential "real" effect. Also, why so low concentrations of Gem and Sel? Why 4-5 days treatment, and not 9-days like they most often did?

Supp. Figure S4B: The "high" and "low" p-mTOR staining criteria are available on the TGCA database? Or the authors came up with a quantitative strategy to decide this? If yes, how?

Supp. Figure S6B: Adavosertib or Chloroquine were added either at the beginning or 5 days after the start. What is the final day of the experiment? How many days after Day 5?

Supp. Figure S7A: Autophagic influx is usually measured by LC3-II / LC3-I ratio. Although the

increase of LC3-II is a strong marker, the appropriate way is to take the ratio.

Supp. Figure S7B: Need to show total CREB, ATF1 and ATG14 levels, especially as they seem to have a loading problem. And again, LC3-II / LC3-I ratio is a more appropriate way to measure autophagic induction than LC3-II alone.

Supp. Figure S7C,D: Was the quantification of LC3 puncta normalized to cell number?

Supp. Figure S7F: Why did they have to use different autophagy inhibitors in PaCa-2 and HeLa cells (MRT68921 vs. Bafilomycin A1)? I can understand using different chemotherapeutic agents, as these are different types of cancers, but why do they constantly have to change all other inhibitors, concentrations and time points without a valid reason?

Supp. Figure S7G: Was Chloroquine added since the beginning or after a certain number of days? They had done both kind of experiments earlier, and now have to be more clear as to what they did.

Supp. Figure S8A: When was Adavosertib added; since the beginning or after a certain number of days?

Supp. Figure S8B: Is it possible to normalize the Y-axes among all individual plot profiles somehow? Or it would be necessary to show the axis intervals for each mini-panel. The plot profiles (heights of the peaks) look misleading in the middle panel compared to the side panels when the quantification values (at the top corners) are taken into consideration. It also seems like there is a stronger polyploidy in Etoposide and Irinotecan treated cells than Gemcitabine and Selinexor. Coincidentally, Eto and Irino cells (high polyploid) seem to have more G2 accumulation. Any comments on this?

	Cell type	Treatment	Treatment	Treatment	Treatment	Treatment	Treatment	Treatment
Figure 1	4292-Cas9	Gemcitabine, 20nM, 12 days	Selinexor, 0.33uM, 12 days					
Figure 2A	4292-Cas9	Gemcitabine, 2uM, 7 days	Selinexor, 2uM, 7 days	Rapamycin, 100nM, 7 days	Everolimus, 100nM, 7 days	INK128, 100nM, 7 days	Torin1, 100nM, 7 days	JR-AB2-011, 1uM, 7 days
Figure 2B	MIA PaCa2	Gemcitabine, 100nM, ? days	Torin1, ? conc., ? days					
	HeLa	Irinotecan, 60uM, ? days	Torin1, ? conc., ? days					
	DU145	Mitoxantrone, 100nM, ? days	Torin1, ? conc., ? days					
	MeWo	Paclitaxel, 10nM, ? days	Torin1, ? conc., ? days					
Figure 2C	MIA PaCa2	Gemcitabine, ? conc.	Torin1, ? conc.					
Figure 2D	MIA PaCa2	? chemotherapy, ? conc., ? days	Torin1, ? conc., ? days + 11 days					
Figure 3B	MIA PaCa2	Gemcitabine, 30nM, 3-6-9 days	Selinexor, 200nM, 3-6-9 days					
Figure 3C	MIA PaCa2	Gemcitabine, ? conc., 6 days	Selinexor, ? conc., 6 days					
Figure 3D	NSC mice+WT or TSC2-/- tumors	Gemcitabine, ? conc., 4 weeks						
Figure 3E	NSC mice+WT or TSC2-/- tumors	Gemcitabine, ? conc., 4 weeks						
Figure 4A,B	MIA PaCa2	Gemcitabine, 100nM, 8 days	Selinexor, 2uM, 8 days	Etoposide, 6uM, 8 days	Irinotecan, 6uM, 8 days	Torin1, 100nM, 8 days		
Figure 4C,D	MIA PaCa2	Gemcitabine, 100nM, 9 days	Torin1, 100nM, 9 days					
Figure 4E	MIA PaCa2	Gemcitabine, ? Conc., 9 days	Selinexor, ? Conc., 9 days	Etoposide, ? Conc., 9 days	Irinotecan, ? Conc., 9 days	Torin1, ? Conc., 9 days		
Figure 4F	MIA PaCa2	Gemcitabine, 100nM, 9 days	Selinexor, 1uM, 9 days	Etoposide, 6uM, 9 days	Irinotecan, 6uM, 9 days	Torin1, ? conc., 9 days		
Figure 5A	MIA PaCa2	Gemcitabine, ? Conc., 9 days	Torin1, ? Conc., 9 days					
Figure 5B	MIA PaCa2	Gemcitabine, 100nM, ? days	Torin1, 100nM, ? days	Adavosertib, 100nM, ? days	CHIR124, 100nM, ? days	Ceralasertib, 10uM, ? days	Choloroquine, 10uM, ? days	Bisindolylm V, 30uM, ? days
	HeLa	Irinotecan, 60uM, ? days	Torin1, 100nM, ? days	PD407824, 3uM, ? days	Prexasertib, 1uM, ? days	Berzosertib, ? Conc., ? days	Choloroquine, 10uM, ? days	LY2584702, 1uM, ? days
Figure 6A	??? (MIA PaCa2 ???)	Gemcitabine, ? Conc., ? days	Selinexor, ? Conc., ? days	Etoposide, ? Conc., ? days	Irinotecan, ? Conc., ? days	Torin1, ? Conc., ? days		
Figure 6B	MIA PaCa2	Gemcitabine, ? Conc.	Torin1, ? Conc.					
Figure 6C	MIA PaCa2	Gemcitabine, 100nM, ? days	Torin1, 100nM, ? days					
Figure 6D	MIA PaCa2	Gemcitabine, 100nM, ? days	Selinexor, 2uM, ? days	Etoposide, 6uM, ? days	Irinotecan, 6uM, ? days	Torin1, ? Conc., ? days		
	Cell type	Treatment	Treatment	Treatment	Treatment	Treatment	Treatment	Treatment
Supp. Figure S2A	MIA PaCa2	Doxorubicin, 100nM, 9 days	Selinexor, 2uM, 9 days	Paclitaxel, 10nM, 9 days	Etoposide, 6uM, 9 days	Mitoxantrone, 60nM, 9 days	Irinotecan, 6uM, 9 days	Torin1, 100nM, 9 days
Supp. Figure S2B	PC3	Paclitaxel, 300nM, 9 days	Torin1, 100nM, 9 days					
	T-47D	Carboplatin, 40uM, 9 days	Torin1, 100nM, 9 days					
	MeWo	Paclitaxel, 10nM, 9 days	Torin1, 100nM, 12 days					
Supp. Figure S3A	MIA PaCa2	Gemcitabine, ? Conc.	Torin1, ? conc.					
Supp. Figure S4A	MIA PaCa2	Gemcitabine, upto 2.5nM, 4-5 days	Selinexor, upto 3.5nM, 4-5 days	5-FU, upto 6nM, 4-5 days	Paclitaxel, upto 1.5nM, 4-5 days			
Supp. Figure S5A	MIA PaCa2	Gemcitabine, ? Conc., 9 days	Torin1, ? conc., 9 days					
Supp. Figure S5B	MDA-Panc-28	Gemcitabine, 10uM, 6 days	Selinexor, 1uM, 6 days	Etoposide, 3uM, 6 days	Irinotecan, 10uM, 6 days	Torin1, ? Conc., 6 days		
Supp. Figure S6A	MIA PaCa2	Selinexor, 2uM, 9 days	Etoposide, 6uM, 9 days	Irinotecan, 6uM, 9 days	Doxorubicin, 100nM, 9 days	Choloroquine, 10uM, 9 days	Adavosertib, 100nM, 9 days	Torin1, 100nM, 9 days
Supp. Figure S6B	MIA PaCa2	Gemcitabine, 100nM, ? total days	Torin1, 100nM, ? total days	Choloroquine, 10uM, ? days	Adavosertib, 100nM, ? days			
Supp. Figure S6C	MIA PaCa2	Gemcitabine, 100nM, ? days	Torin1, 100nM, ? days	Bafilomycin A1, 2nM, ? days	MHY1485, 20uM, ? days	ULK101, 20uM, ? days	Berzosertib, 100nM, ? days	SPS81, 300nM, ? days
					SRA737, 3uM, ? Days	Abemaciclib, 600nM, ? days	GSK2334470, 3uM, ? days	LJI308, 30uM, ? days
Supp. Figure S7A	MIA PaCa2, PANC-1	Torin1, 100nM, upto 19 days						
Supp. Figure S7B	MIA PaCa2	Gemcitabine, 100nM, ? days	Selinexor, 2uM, ? days	Etoposide, 6uM, ? days	Irinotecan, 6uM, ? days	Torin1, ? Conc., ? days		
Supp. Figure S7C,D	MIA PaCa2	Gemcitabine, ? Conc., ? days	Selinexor, ? Conc., ? days	Etoposide, ? Conc., ? days	Irinotecan, ? Conc., ? days	Torin1, ? Conc., ? days		
Supp. Figure S7F	MIA PaCa2	Gemcitabine, 100nM, ? days	Torin1, 100nM, ? days	SAR405, 10uM, ? days	MRT68921, 5uM, ? days			
	HeLa	Irinotecan, 60uM, ? days	Torin1, 100nM, ? days	SAR405, 10uM, ? days	Bafilomycin A1, 3nM, ? days			
Supp. Figure S7G	MDA-MB-231	Paclitaxel, 10nM, ? days	Torin1, 100nM, ? days	Choloroquine, ? Conc., ? days				
	DU145	Paclitaxel, 10nM, ? days	Torin1, 300nM, ? days	Choloroquine, ? Conc., ? days				
	SW480	Gemcitabine, 1uM, ? days	Torin1, 300nM, ? days	Choloroquine, ? Conc., ? days				
Supp. Figure S8A	MDA-MB-231	Paclitaxel, 10nM, ? days	Torin1, 100nM, ? days	Adavosertib, ? Conc., ? days				
	DU145	Paclitaxel, 10nM, ? days	Torin1, 300nM, ? days	Adavosertib, ? Conc., ? days				
	SW480	Gemcitabine, 1uM, ? days	Torin1, 300nM, ? days	Adavosertib, ? Conc., ? days				
Supp. Figure S8B	MIA PaCa2	Gemcitabine, 100nM, 9 days	Selinexor, 2uM, 9 days	Etoposide, 6uM, 9 days	Irinotecan, 6uM, 9 days	Torin1, ? Conc., 9 days		

Reviewer #3:

Remarks to the Author:

In their manuscript, the authors describe the identification of the mTOR pathway as a mediator of the response to chemotherapy. They find a strong enrichment of mTOR-related genes in their CRISPR screen and subsequently demonstrate that suppression of the mTOR pathway leads to chemotherapy resistance while activation increases chemosensitivity. Most interestingly, mTORi appears to promote survival of a subpopulation of cells ('persisters') by inducing a reversible form of senescence associated with G2/M arrest and the activation of autophagy. The experiments are well done and importantly, their findings suggest that therapeutic combinations that seek to use chemotherapeutics with mTORi should be avoided or evaluated carefully due to the potential for great resistance.

MAJOR COMMENTS:

- Is there quantification of the data shown in Figure 2A/B (and also Fig 5B)? In addition to the images, it would be informative to include a quantification of the cell number/density in each condition. With the images alone, it is difficult to ascertain the fraction of cells which become 'persisters'. Are these highly rare cells or a less rare subpopulation?
- In Figure 2A/B, the authors do an excellent job using multiple inhibitors to tease apart the role of mTORC1 versus mTORC2. But what are the effects of these mTOR inhibitors alone on these cells? Mainly, what is the effect of Torin1 alone? Does it induce any response or effect on cell survival/proliferation?
- In the RNA-seq experiment for Figure 2D, how many genes are upreg./downreg. in the mTORi alone condition? This condition is alluded to but not shown or described. The main question I have is whether the authors think the gene expression change seen in persister cells is being driven mainly by the mTOR inhibition or the combination of chemo+mTORi? This is important to establish as the authors conclude (line 138 of main text) "The acquired resistance following mTOR inhibition is therefore unlikely mediated by stable genetic changes, but a consequence of an adaptive response to chemotherapeutic stress." But from looking at Fig S3B, much of the signatures are shared between the mTORi and persister conditions, suggesting that the resistance may be more likely due to a response to mTORi rather than due to "chemotherapeutic stress"?
- In Figure 2E, the authors cite several published studies of "residual tumors following chemotherapy"; however the colorectal cancer study (ref 18, Lupo et al) does not seem to use any chemotherapies and instead involves a targeted therapy against EGFR, this distinction is important. Interestingly, the association with the MP gene signature appears to be best correlated with this data set.
- It is remarkable that mTORi induces persister cells across so many cell lines and types of chemotherapies. As the authors mention, there is a strong correlation between TP53 status and whether mTORi induces persisters. Have the authors tried knocking down/out TP53 in a wild-type cell line to determine whether this is sufficient to enable cells to become persisters?
- Have the authors tested whether the chemo+mTORi persister phenomenon they observe occurs in vivo? I recognize that this may be challenging to demonstrate, especially if chemotherapy alone in vivo is unable to completely eradicate the tumor cells. Alternatively, if chemotherapy alone in vivo does result in residual tumor cells and resistance, is this because in vivo conditions modify mTOR signaling in the tumor cells (even in the absence of mTORi) compared to in vitro conditions?
- For Figure S4A, the IC50 values should also be shown to help interpret the data. Also, for the paclitaxel treatment, more doses are recommended to generate a more accurate dose response curve.

MINOR COMMENTS:

- In Figure 1D, what are the units of the heatmap scale?, this is not labeled. Also, how were these genes ranked in the heatmap? This is not specified in the figure legend or methods. Additionally, I am curious to know what was the ranking of components of the mTORC2 complex in the CRISPR screen (Rictor, etc), as their data suggest the persister state is only dependent on mTORC1.
- How were the genes ranked in Figure S1? The details of ranking are not found in the figure legend or methods section.
- For Figure 5A, the text mentions compounds targeting "cell death pathways (apoptosis and ferroptosis)", but these are not labeled as a distinct category in the figure.

- For Figure 6, the quantification of cell cycle based on the FUCCI reporter should be shown for the additional conditions in panel A (and not just for Gem + T)

REVIEWER COMMENTS

Reviewer #1 (Remarks to the Author):

The manuscript by Liu et al. uses functional genomics to identify genes conferring resistance or sensitivity to chemotoxic agents in cancer cells. They screen a genome-wide CRISPR knockout library in p53/K-Ras mutant cells treated with chemotoxic agents. They reveal that the loss of function mutations in the mTOR pathway favors the persistence of a pool of senescent-like cells that can re-grow in favorable conditions. The data are interesting and well controlled. The novelty is somehow limited, as several studies already pointed that mTOR inhibition may be chemoprotective and radioprotective (including papers by A. Amon's lab not cited). More information would be required to better understand the precise cellular adaptations to these chemotoxic and genetic treatments.

We thank Reviewer #1 for the insightful and constructive comments. While mTOR inhibition has been reported to be chemoprotective and radioprotective, our study is the first to systematically investigate the cytoprotective effects across multiple cancer types and reveal the mechanistic basis underlying the protection. In the revised manuscript, we have cited more relevant studies^{1,2}, as well as expanded our discussion section.

1) After the genetic screen (Figure 1), are all the mutations homozygous or some are heterozygous? It would be surprising if the homozygous knockout of Rps6 and mTOR maintained viable cells.

This is an interesting question frequently raised in CRISPR screening studies. Genomic changes generated in CRISPR screens are highly heterogeneous with a mixture of both homozygous and heterozygous mutations depending on the function of the specific genes. In the case of RPS6 and mTOR, the mutations are likely various hypomorphic mutations as their total loss of function may not be compatible with viability.

2) Kinetics analysis at early time points should be provided after chemotoxic treatment alone (Gem and Sel, Figure 2). Are some treated cells undergoing senescence? Or exclusively apoptosis? Is Torin1 treatment shifting from apoptosis to senescence?

Cell survival data following treatments were shown in Figure 2C. To examine the kinetics of apoptosis and senescence, we performed senescence-associated beta-galactosidase (SA- β -gal) staining and the caspase 3/7 activity assay at days 1, 3, and 5 following treatments. The findings are shown below. Cells treated with chemotherapeutic agents alone (100nM Gemcitabine and 2 μ M Selinexor) underwent both senescence and apoptosis. The addition of Torin1 treatment partially reduced apoptosis without affecting the cellular senescence induced by chemotherapeutics. Therefore, senescence is primarily due to the chemotherapeutic agents while mTOR inhibition supports survival without significant influence on cellular senescence.

3) Similarly, kinetics analysis at early time points should be provided during the recovery phase. How is this reversible senescence characterized? Do all the persisters cells recover? Or just a sub-population? Some kinds of single cell tracing should be assessed.

The recovery phase was shown in Fig 2C. Additionally, following the reviewer's suggestion, we performed single cell tracing upon persister recovery in MIA PaCa-2 cells expressing the FUCCI reporter. The red color represents the G1/S while the green color indicates S/G2/M phase of the cell cycle. Comparing the same imaging fields for persisters, and 3 and 7 days post-recovery, we observed that approximately 20% of persisters recovered and resumed proliferation to form single-cell colonies, 50% underwent cell death, and the remaining 30% stayed arrested.

4) Is it possible that Cdk4 and Cdk6 inhibitors protect because the cells arrest in G1 phase and are not exposed to the replication stress of the chemotoxic agent? Or do the cells treated with Cdk4/6 inhibitors also accumulate in G2 phase?

Cdk4/6 inhibitors have been reported to arrest cells in G1 phase^{3,4}. We examined the cell cycle distribution by propidium iodide (PI) staining in cells treated with Cdk4/6 inhibitor (abemaciclib, 600nM) alone, or in combination with 100nM gemcitabine for nine days. In line with previous reports, abemaciclib as a single agent increased the cell number in G0/G1 phase but not G2/M, whereas co-treatment induced persisters that accumulated in the G2/M phase and polyploid state.

How is it possible that torin1 treatment arrests cells in G1 but the combined Gem + Torin treatment in G2?

mTOR inhibition is known to increase cell number in the G1 phase. However, it also significantly downregulates the expression of genes essential to G2/M transition, such as PLK1, FOXM1, MYBL2, and Cyclin B1 (Fig 6). The G2/M arrest is primarily induced by the gemcitabine treatment. mTOR inhibition further potentiates the G2/M arrest and delays the G2/M transition. Therefore, combined gemcitabine and Torin1 treatment leads to significant G2/M arrest.

Does the same happen with S6K inhibitors? The S6K inhibitors should not affect cell cycle progression but rather cell size.

S6K inhibition behaves similarly to mTOR inhibition. S6K inhibition alone with 30 μ M Bisindolylmaleimide V for nine days increases the cell number in G0/G1 phase. In contrast, combined treatment with Bisindolylmaleimide V and 100nM gemcitabine increases the number of cells in G2/M phase and polyploid state (top panel).

Based on the FSC/SSC data in flowcytometric analysis, Bisindolylmaleimide V treatment alone moderately increases the cellular granularity (SSC), while the combination with gemcitabine increases both cell size and granularity (bottom panel).

Reviewer #2 (Remarks to the Author):

Liu et al present data supporting the notion that mTORC1 inhibition leads to increased autophagy and a G2/M cell cycle arrest and that this facilitates a subpopulation of tumor cells to survive chemotherapy allowing subsequent regrowth.

Their observations are interesting and thought provoking. However, they also stand in the face of an enormous number of publications suggesting that mTORC1 inhibition in combination with other anti-cancer chemotherapeutics is a valid treatment strategy. The authors should discuss this disconnect in the Discussion.

We thank Reviewer #2 for the extensive and constructive comments.

While some preclinical studies suggest combining mTORC1 inhibition and chemotherapeutics as a valid treatment strategy, it has not been replicated in clinical trials. According to the NIH clinical trials database, mTOR inhibitors have been tested in over 500 trials over the past decades. Among approximately 200 completed trials with mTOR inhibitors as a monotherapy or part of a combinational therapy, the vast majority have shown modest to no benefit. Not surprisingly, the FDA approval of mTOR inhibitors so far has been limited to the control of late-stage tumor expansion in a few specific cancer types, including renal cell carcinoma, pancreatic neuroendocrine tumors, and hormone positive HER2-negative breast cancer in postmenopausal patients.

The discrepancy may be partially due to the difference in the readouts. Preclinical studies often rely on short-term delayed tumor expansion as the readout of treatment outcome. However, reduced residual tumor

burden, but not the short-term delayed tumor expansion, is the key determinant of clinical benefit and patients' survival.

Our study attempts to raise awareness on the potential adverse effects of combining DNA damage-inducing agents and mTOR inhibitors for cancer therapy, and emphasize the need to carefully evaluate the benefits of treatment regimens involving mTOR inhibitors in clinical trials. We have revised the Discussion section accordingly:

“The mTOR pathway is widely deemed as an appealing cancer therapeutic target. However, the expectation has not been met in clinical trials. Among more than 500 trials using mTOR inhibitors as a single agent or in combination with other drugs, few have demonstrated clinical benefit. Not surprisingly, FDA approval of mTOR inhibitors, including everolimus and temsirolimus, is only limited to the control of late-stage tumor expansion in a few specific cancers. We observe that mTOR suppression promotes survival of various cancer cells undergoing chemotherapy and dampens therapeutic efficacy. Similarly, others have shown that mTOR inhibition is chemoprotective in leukemia cells. Thus, the use of mTOR inhibitors can be counterproductive in the context of chemotherapy. This may explain the general lack of efficacy of mTOR inhibitors in cancer clinical trials and argues against their widespread use in combination with DNA damage-inducing agents. In this light, stimulation of mTOR signaling, but not its inhibition, may be the strategy of choice to reduce residual diseases and improve chemotherapeutic efficacy.”

General comments:

For figures containing microscopic images, scale bars should be defined in the figure legend. For figures containing color bars, it may be better to indicate whether it is fold change or $\log_2(\text{fold change})$ on the figure rather than in the figure legend for ease. Figure legends were often missing crucial information, such as concentrations of all compounds used or time points of the experiments (all marked in the excel sheet).

Although we do not have access to the excel sheet mentioned by Reviewer #2, we have revised the figures and figure legends according to the detailed comments.

They used quite a high number of chemotherapeutic agents or other small molecule inhibitors on a variety of cells. How did they decide the working concentration for all these compounds per cell type? Did they make preliminary experiments with variable concentrations and landed on these? If yes, how were they tested, what were the selection criteria? If no, do they have solid evidence from literature that these are the “standard” concentrations widely used in the field for each cell type?

For the persister-induction studies in Fig 2 and Fig S2, we performed titration experiments to identify the lowest concentrations of chemotherapeutics that would eradicate the tumor cells within seven days, before examining the protective effects of mTOR inhibitors. We have updated the Methods section with the details.

For the small-molecule chemical library screen, we tested multiple concentrations for each inhibitor (1:3 dilution from 10 μ M to 100nM). In cases where these concentrations did not apply, concentrations as low as 30nM, or as high as 100 μ M were tested. Each concentration was simultaneously tested on the parental cells and persisters. Concentrations leading to significant changes in viability of either parental cells or persisters were tabulated, and only the lowest concentration was reported in Fig 5 and Table S6. In cases where both parental cells and persisters were eradicated, the experiment was repeated using lower concentrations. For inhibitors reported in multiple studies, we consulted with the original publications for the typical concentration ranges and revised our designs accordingly. We have updated the Methods section with the technical details.

Although they have acceptable reasons to use different concentrations of a certain compound in different experiments (like Crispr screen vs. other cell culture experiments), there are identical experiments having used

different concentrations of the same compound, or more commonly, different time points! There is very low temporal consistency among persister-cell generating experiments with no given explanation.

For our persister model based on MIA PaCa-2 cells, we have consistently used 100nM Torin1 and 100nM gemcitabine for nine days for the vast majority of experiments, including RNA-seq, C12-FDG assay, Western Blot analysis of senescence markers (Fig 4C-E), small molecular inhibitor screen (Fig 5), FUCCI experiments (Fig 6), Western Blot of G2/M proteins (Fig 6), Western Blot analysis of autophagy markers (Fig S7B), as well as LC3 reporter assay (Fig S7C-D). We have specified the details in the figure legends. For the SASP Luminex cytokine assay, the media was changed on day seven and collected on day nine. The SABG staining was performed on cells treated for eight instead of nine days due to logistical reasons (Fig 4). For the persister-generating experiments using various cell lines and chemotherapeutic drugs, we have generally used seven to nine days of treatment. We have included the technical details in Table S2.

They show no biochemical proof for the inhibitors they used (i.e. phospho-western blots etc.). Did they test the inhibition efficiency of said compounds at the given concentrations and time points?

We have provided the biochemical evidence on the expected functionality of the mTOR, WEE1, CHK1, ATR, and S6K inhibitors. The Western blot data were provided in Figure S2 and S6 of the revised manuscript and shown below.

They do not mention the number of replicates for most of the experiments. How many times did they replicate the “Persister cell generation +/- various compounds” experiments? They should include this information in EVERY figure legend.

Experimental detail is lacking – see Xlsx file

Experiments involving the MIA PaCa-2 persister model were performed with at least three biological replicates, with the exception of the small-molecule chemical library screen, which used two biological replicates. Experiments using other cell lines were performed with at least two biological replicates. The information has been updated in the figure legends.

Specific comments (underlined text = experiments need to be performed):

Figure 2A: They show no validation of the inhibition efficiency for any of the compounds. They should show via phospho-immunoblots that 1) the compounds inhibit mTORC1 and/or mTORC2 in untreated cells at various concentrations spanning their final working concentration, and 2) mTORC1 (and also maybe mTORC2) activity is reduced in Chemo+mTORi samples compared to untreated or recovered cells. Likewise, they can show mTORC1/2 re-activation in recovering cells in a time course.

We have examined the efficiency of mTOR inhibitors and presented the data in Fig S2 as well as below. We use pS6K-S371 and p4EBP1-S65 as the indicators for mTORC1 activity, and pAKT-S473 for mTORC2 activity.

In persisters, we observed decreased pS6K-S371 and p4EBP1-S65, which rebound following drug withdrawal. In contrast, pAKT-S473 activity was upregulated in persisters and downregulated following drug withdrawal. This data show that the persisters have decreased mTORC1 but not mTORC2 activity.

Figure 2D: Figure legend says “Black dots represent differentially ...”. Those dots in the figure are blue.

The figure legend has been corrected.

Figure 3B: Having different colors for WT and TSC mutants would make it easier and quicker to understand. They can match the colors with Supp. Figure S4A.

The y-axis represents the ratio of each TSC KO clone to the wild type clone in the multicolor competition assay. We aim to monitor changes in the ratios over time with or without chemotherapy. Therefore, it would be difficult to color-code the knockout vs wild type clones. However, we did revise and color-code the y-axis legend to avoid the confusion.

Figure 3D: What were the initial sizes of the WT vs TSC2-KO tumors? Fold change size difference may be misleading, as it is processed data. It is necessary (or better scientific practice) to present all four data sets: WT and TSC2-KO sizes before and after Gemcitabine treatment. It would be also good, if they could provide representative images from BLI or MRI, showing tumor sizes before and after treatment.

The initial tumor size in the WT vs TSC2 KO cohort was similar (Student's t-test, $p = 0.1225$) based on BLI quantifications. In the revised manuscript, we have included the pre- and post-treatment BLI quantification for the WT and TSC KO tumors in Table S5. The representative BLI images are included in Fig S4 and shown below.

Figure 3E: They MUST include survival curves of WT and TSC2-KO xenograft injected mice without treatment. How do we know the effect of Gemcitabine treatment? What if TSC2-KO tumor mice already have a better survival???

The survival curve of mice carrying WT and TSC2 KO tumors without treatments (dashed lines) are included in the revised Fig 3E and shown below. Untreated WT and TSC2 KO groups do not show significant difference in survival.

Figure 5A: It may be useful to add "% survival" to the axes titles to understand easily. They did not mention which cell line was used for this experiment in the figure legend.

We have added “% survival” in the axis titles. We have also included the cell line information in the figure legend.

Figure 5B: Figure legend has Adavosertib treatment performed to HeLa cells, however the figure does not contain this treatment. Instead there is Benzoseritib treatment which is not mentioned in the figure legend.

It should be the Berzosertib instead of Adavosertib in the legend. We have corrected accordingly.

Like mTORC1/2 inhibitors, they should validate the inhibition capacity of their WEE1, CHK1 and ATR inhibitors at given concentrations via phospho-immunoblots.

We have validated the inhibition capacity of the WEE1, CHK1, ATR, and S6K inhibitors at the given concentrations. As WEE1 phosphorylates CDC2 on tyrosine 15, we used pCDC2-T15 as a marker for WEE1 activity. Interestingly, we also observed the paradoxical accumulation of S345-phosphorylated CHK1 following inhibition of CHK1 kinase activity, as reported previously by others⁵.

Figure 6A,B: Do they have quantification data from the other treatments than Gem+T? Are there G2/M ratio differences among these chemotherapeutic drugs? Does it correlate with the flow cytometry analyses in Supp. Figure 8B, and the difference in polyploidy & G2 accumulation?

We have included the quantification data for other treatments in Fig 6B. Gemcitabine, irinotecan, and etoposide induce pronounced G2/M arrest, while selinexor has a modest effect. The data is in agreement with the flow cytometric analysis shown in Fig S8.

Figure 6D: Total CHK1 levels need to be shown.

The total levels of CHK1 have been added in Fig 6D.

Supp. Figure S2A: Figure and figure legend indicates Selinexor, but main text says Gemcitabine (line 108).

We have corrected the main text.

Supp. Figure S2B: This panel is not related to where it is mentioned in the text (line 117). Actually, there is no proper explanation of this panel in the Results.

The original Fig S2B is redundant and we have removed it in the revision.

Supp. Figure S2C: How consistent were the experimental conditions among all these treatments? Same

concentration and time points were used? What was the “yes” or “no” selection criteria, the lack of which led to “not determined” phenotype?

For experiments listed in Fig S2F, we tested the chemotherapeutic agents in each cell type at 1:3 dilution to determine the minimal killing concentrations at day 7 of treatment. Subsequently, tumor cells were treated with chemotherapeutic agents at the minimal killing concentration and Torin1 (100nM) for 7-9 days to induce persisters. We have updated the figure legends and provided the details in Table S2 (also shown below). Additionally, we have updated the notes as below:

“+”: Persisters induced by treatment with the chemotherapeutic agents at the minimal killing concentrations plus 100nM Torin1.

“-”: No persisters observed following combined treatment with chemotherapeutic agents and 100nM Torin1.

“n.d.”: Not determined due to intrinsic resistance to the specific chemotherapeutic agents.

Cancer types	Cell lines	TP53 status	Chemotherapeutic agents	Concentrations tested for each chemotherapeutic agent	Minimal killing concentrations	Persister phenotype	Timeline	
Pancreatic cancer	MDA-Panc-28	Mutant	Gemcitabine	10nM-100µM	10µM	+	7d	
			Irinotecan	300nM-30µM	3µM	+	7d	
			Etoposide	300nM-30µM	1µM	+	7d	
	BxPC-3	Mutant	Gemcitabine	3-300nM	200nM	+	8d	
			Irinotecan	300nM-30µM	3µM	+	8d	
			Etoposide	300nM-30µM	3µM	+	8d	
	MPanc96	Mutant	Gemcitabine	300nM-30µM	30µM	-		
			Irinotecan	300nM-30µM	30µM	+	9d	
			Etoposide	300nM-30µM	30µM	+	9d	
	Capan-2	WT	Gemcitabine	3-300nM	100nM	+	9d	
			Irinotecan	300nM-30µM	30µM	-		
			Etoposide	300nM-30µM	30µM	-		
Prostate cancer	LNCaP	Mutant	Paclitaxel	1-100nM	30nM	+	7d	
			Mitoxantrone	1-100µM	100µM	-		
			Estramustine	10nM-10µM	10µM	-		
	C4-2B	Mutant	Paclitaxel	1-100nM	30nM	+	9d	
			Mitoxantrone	10nM-1µM	300nM	+	15d	
			Estramustine	1-100µM	30µM	+	9d	
	DU 145	Mutant	Paclitaxel	1-100nM	10nM	+	7d	
			Mitoxantrone	10nM-1µM	300nM	+	7d	
			Estramustine	1-100µM	100µM	+	7d	
	22Rv1	Mutant	Paclitaxel	1-100nM	30nM	+	7d	
			Mitoxantrone	10nM-1µM	1µM	-		
			Estramustine	1-100µM	100µM	+	7d	
	PC3	Mutant	Paclitaxel	1nM-1µM	300nM	+	9d	
			Mitoxantrone	10nM-1µM	1µM	-		
			Estramustine	1-100µM	n.d.	n.d.		
	Breast cancer	T-47D	Mutant	Paclitaxel	1nM-30µM	10µM	+	9d
				Carboplatin	30nM-40µM	30µM	+	9d
				Gemcitabine	100nM-10µM	300nM	+	9d
MCF7		WT	Paclitaxel	1-100nM	30nM	-		
			Carboplatin	10nM-30µM	30µM	-		
			Gemcitabine	100nM-10µM	1µM	-		
MDA-MB-231		Mutant	Paclitaxel	1-100nM	10nM	+	9d	
			Carboplatin	1-100µM	100µM	+	9d	
			Gemcitabine	100nM-10µM	1µM	+	9d	
SK-BR-3		Mutant	Paclitaxel	1-100nM	10nM	+	9d	
			Carboplatin	1-100µM	50µM	+	14d	
			Gemcitabine	10nM-10µM	300nM	+	9d	
Liver cancer	PLC/PRF/5	Mutant	Gemcitabine	10nM-1µM	100nM	+	9d	
			Doxorubicin	30nM-3µM	100nM	+	9d	
			Mitoxantrone	10nM-10µM	100nM	+	9d	
	SNU398	Mutant	Gemcitabine	10nM-1µM	30nM	+	9d	
			Doxorubicin	10nM-10µM	30nM	+	9d	
			Mitoxantrone	10nM-10µM	300nM	+	9d	
	Hep3B	Mutant	Gemcitabine	10nM-10µM	10µM	-		
			Doxorubicin	10nM-10µM	3µM	-		
			Mitoxantrone	10nM-10µM	1µM	-		
	HepG2	WT	Gemcitabine	30nM-3µM	300nM	-		
			Doxorubicin	30nM-3µM	1µM	-		
			Mitoxantrone	10nM-30µM	30µM	-		
Melanoma	Mewo	Mutant	Paclitaxel	1nM-30µM	10nM	+	9d	
			Carboplatin	1-100µM	30µM	+	9d	
			Gemcitabine	10nM-30µM	3µM	+	9d	
	A375	WT	Paclitaxel	1nM-300nM	10nM	-		
			Carboplatin	1-100µM	n.d.	n.d.		
			Gemcitabine	1nM-1µM	1µM	-		
NM2C5	Mutant	Paclitaxel	1-100nM	10nM	+	9d		
		Carboplatin	1-100µM	100µM	+	12d		
		Gemcitabine	30nM-10µM	3µM	+	9d		
Colon cancer	HCT116	Mutant	Irinotecan	1-100µM	n.d.	n.d.		
			Gemcitabine	100nM-10µM	3µM	-		
			Etoposide	1-100µM	30µM	-		
	SW480	Mutant	Irinotecan	1-100µM	3µM	+	9d	
			Gemcitabine	100nM-10µM	1µM	+	9d	
			Etoposide	1-100µM	3µM	+	9d	
Lung cancer	H460	WT	Etoposide	30nM-30µM	30µM	-		
			Gemcitabine	10nM-1µM	1µM	-		
			Irinotecan	30nM-30µM	10µM	-		
	A549	WT	Etoposide	1-100µM	100µM	-		
			Gemcitabine	10nM-30µM	10µM	-		
			Irinotecan	1-100µM	30µM	-		
	H1299	Mutant	Etoposide	1-100µM	30µM	+	9d	
			Gemcitabine	10nM-10µM	3µM	+	9d	
			Irinotecan	1-100µM	10µM	+	9d	
	H441	Mutant	Etoposide	1-100µM	30µM	+	7d	
			Gemcitabine	10nM-10µM	300nM	+	7d	
			Irinotecan	1-100µM	30µM	+	7d	
Head and neck cancer	SCC47	WT	Paclitaxel	1-100nM	100nM	-		
			Carboplatin	1-100µM	n.d.	n.d.		
			Gemcitabine	10nM-10µM	1µM	-		
Ovarian cancer	SK-OV-3	Mutant	Etoposide	1-100µM	10µM	+	9d	
			Gemcitabine	10nM-10µM	3µM	+	9d	
			Irinotecan	1-100µM	30µM	+	9d	
Cervical cancer	HeLa	Mutant	Etoposide	1-100µM	10µM	+	7d	
			Gemcitabine	10nM-10µM	n.d.	n.d.		
			Irinotecan	1-100µM	60µM	+	7d	
Osteosarcoma	U2OS	WT	Etoposide	300nM-30µM	1µM	+	9d	
			Gemcitabine	30nM-3µM	300nM	+	9d	
			Doxorubicin	10nM-1µM	100nM	+	9d	

Supp. Figure S3B: The axes may be labeled on the figure (probably “NES”..?). The top graph shows pathway dysregulations between Torin1 treated and control cells, and it seems to have a similar dysregulation pattern compared to persister vs. control cells graph. First of all, what are Torin1 samples? The RNA-seq was performed with Control, Persister and Recovery cells. Did they have an additional “only Torin1 treated” cells next to Persisters? Then, why are the expression profile of Torin1 samples is almost the same as Persisters? The associated text in the Results (lines 128-134), does not clearly explain this top panel. Lower panel is clear.

We have labeled the axes with NES in the revised figures. We have performed RNA-seq analysis of four groups of cells: control, Torin1-treated, persisters, and recovery. Torin1 vs control and persister vs control groups have similar GSEA profiles based on the 50 hallmark signatures, which highlights the role of mTOR-regulated transcriptional change in persisters. We have updated the main text and included the RNA-seq analysis of Torin1-treated cells in Table S3. Although the transcriptomes of mTORi-treated cells and persisters have some similarity, they aren't identical. Senescence classification based on the RNA-seq data clearly shows a senescence profile in the persisters but not the Torin1-treated cells (Fig. 4G).

Supp. Figure S4A: Having different colors for WT and TSC mutants would make it easier and quicker to understand. In contrast to their claim, 5-FU and Paclitaxel treated cells do not seem to have an increased sensitivity. Same goes for Gemcitabine. Only Selinexor seems potentially significant. They should have done appropriate statistical analyses to be able to conclude that. The figure legend is missing info: the error bars are showing SD or SEM? In any case, judging from the variation, only Selinexor looks like a potential “real” effect. Also, why so low concentrations of Gem and Sel? Why 4-5 days treatment, and not 9-days like they most often did?

Our study focuses on the persister phenotype generated following nine days of treatment. IC50 experiments typically performed on a 3-5 day timeline may not be as meaningful. Therefore, we have removed the IC50 profiles in the original Fig S4A.

Supp. Figure S4B: The “high” and “low” p-mTOR staining criteria are available on the TGCA database? Or the authors came up with a quantitative strategy to decide this? If yes, how?

Survival analysis of patients' samples were based on the reverse-phase protein array (RPPA) data on phospho-mTOR-S2448 in The Cancer Genome Atlas (TCGA) project. Data were compiled by others using the TRGAted application with pre-determined “high” vs “low” expression⁶. Their compiled data are accessible:

<https://nborcherding.shinyapps.io/TRGAted/>

Supp. Figure S6B: Adavosertib or Chloroquine were added either at the beginning or 5 days after the start. What is the final day of the experiment? How many days after Day 5?

The final day of experiment is day 9, four days after day 5.

Supp. Figure S7A: Autophagic influx is usually measured by LC3-II / LC3-I ratio. Although the increase of LC3-II is a strong marker, the appropriate way is to take the ratio.

Western blots showing both the LC3-II and LC3-I bands have been added to Fig S7A.

Supp. Figure S7B: Need to show total CREB, ATF1 and ATG14 levels, especially as they seem to have a loading problem. And again, LC3-II / LC3-I ratio is a more appropriate way to measure autophagic induction than LC3-II alone.

The total CREB and ATG14 have been added to Fig S7B. pATF1 is not a marker of autophagy. It is detected due to the commonly observed cross-reactivity of the pCREB antibodies. Additionally, we have updated the LC3 blots to show both LC3-II and LC3-I.

Supp. Figure S7C,D: Was the quantification of LC3 puncta normalized to cell number?

Yes, the quantification of LC3 puncta was normalized to cell number.

Supp. Figure S7F: Why did they have to use different autophagy inhibitors in PaCa-2 and HeLa cells (MRT68921 vs. Bafilomycin A1)? I can understand using different chemotherapeutic agents, as these are different types of cancers, but why do they constantly have to change all other inhibitors, concentrations and time points without a valid reason?

Both MRT68921 and Bafilomycin A1 worked well in the two cell lines. We showed different drugs for different cell lines with the intention to demonstrate the generality of the phenomenon. To reduce the confusion, we have removed this figure in the revised manuscript.

Supp. Figure S7G: Was Chloroquine added since the beginning or after a certain number of days? They had done both kind of experiments earlier, and now have to be more clear as to what they did.

Supp. Figure S8A: When was Adavosertib added; since the beginning or after a certain number of days?

Chloroquine and adavosertib were added at the start of the experiment. The figure legend has been updated accordingly.

Supp. Figure S8B: Is it possible to normalize the Y-axes among all individual plot profiles somehow? Or it would be necessary to show the axis intervals for each mini-panel. The plot profiles (heights of the peaks) look misleading in the middle panel compared to the side panels when the quantification values (at the top corners) are taken into consideration. It also seems like there is a stronger polyploidy in Etoposide and Irinotecan treated cells than

Gemcitabine and Selinexor. Coincidentally, Eto and Irino cells (high polyploid) seem to have more G2 accumulation. Any comments on this?

The axis intervals for each mini-panel have been added to the Fig S8. All four drugs display increased cell number in the G2/M phase and polyploid state, with irinotecan and etoposide having more prominent effects. The high level of G2/M arrest and polyploidy may be due to the suppression of the topoisomerase function by irinotecan (a topoisomerase I inhibitor) and etoposide (a topoisomerase II inhibitor).

Reviewer #3 (Remarks to the Author):

In their manuscript, the authors describe the identification of the mTOR pathway as a mediator of the response to chemotherapy. They find a strong enrichment of mTOR-related genes in their CRISPR screen and subsequently demonstrate that suppression of the mTOR pathway leads to chemotherapy resistance while activation increases chemosensitivity. Most interestingly, mTORi appears to promote survival of a subpopulation of cells ('persisters') by inducing a reversible form of senescence associated with G2/M arrest and the activation of autophagy. The experiments are well done and importantly, their findings suggest that therapeutic combinations that seek to use chemotherapeutics with mTORi should be avoided or evaluated carefully due to the potential for great resistance.

We appreciate the constructive and insightful comments from Reviewer #3.

MAJOR COMMENTS:

- Is there quantification of the data show in Figure 2A/B (and also Fig 5B)? In addition to the images, it would be informative to include a quantification of the cell number/density in each condition. With the images alone, it is difficult to ascertain the fraction of cells which become 'persisters'. Are these highly rare cells or a less rare subpopulation?

These persisters generally appear at low frequency (1-5%), but are not highly rare. We have included the cell numbers in the images in Fig 2A-B, 5B, S2E, S2H, and S6.

- In Figure 2A/B, the authors do an excellent job using multiple inhibitors to tease apart the role of mTORC1 versus mTORC2. But what are the effects of these mTOR inhibitors alone on these cells? Mainly, what is the effect of Torin1 alone? Does it induce any response or effect on cell survival/proliferation?

mTOR inhibition mainly has cytostatic effects. Torin1 treatment alone does not induce cell death at concentrations below 10 μ M. The typical nine day Torin1 treatment at 100nM increases cell number in the G0/G1 phase without significant cell death.

- In the RNA-seq experiment for Figure 2D, how many genes are upreg./downreg. in the mTORi alone condition? This condition is alluded to but not shown or described. The main question I have is whether the authors think the gene expression change seen in persister cells is being driven mainly by the mTOR inhibition or the combination of chemo+mTORi? This is important to establish as the authors conclude (line 138 of main text) "The acquired resistance

following mTOR inhibition is therefore unlikely mediated by stable genetic changes, but a consequence of an adaptive response to chemotherapeutic stress." But from looking at Fig S3B, much of the signatures are shared between the mTORi and persisters conditions, suggesting that the resistance may be more likely due to a response to mTORi rather than due to "chemotherapeutic stress"?

RNA-seq analysis of Torin1-treated cells revealed 472 upregulated and 746 downregulated genes with at least a 2-fold change and an adjusted p value of less than 0.01.

We agree with the reviewer that many of the signatures were shared between the mTORi-treated cells and persisters, which confirms a key role of mTORi in the induction of persisters. We have updated the main text accordingly:

"The acquired resistance is therefore unlikely mediated by stable genetic changes, but a consequence of an mTOR inhibition-induced adaptive response to chemotherapeutic stress".

Additionally, although the transcriptomes of mTORi-treated cells and persisters have some similarity, they aren't identical. Senescence classification based on the RNA-seq data clearly shows a senescence profile in the persisters but not the mTORi-treated cells (Fig. 4G).

- In Figure 2E, the authors cite several published studies of "residual tumors following chemotherapy"; however the colorectal cancer study (ref 18, Lupo et al) does not seem to use any chemotherapies and instead involves a targeted therapy against EGFR, this distinction is important. Interestingly, the association with the MP gene signature appears to be best correlated with this data set.

Thanks for pointing out. We have replaced the colorectal cancer study with two reports on rectal and esophageal cancers involving neoadjuvant chemoradiotherapy^{7,8}. In parallel to the gemcitabine/selinexor screens reported in this manuscript, we have also performed the screen in the context of targeted therapy. The role of mTOR in modulating response to targeted therapy is currently being pursued in a separate project.

- It is remarkable that mTORi induces persister cells across so many cell lines and types of chemotherapies. As the authors mention, there is a strong correlation between TP53 status and whether mTORi induces persisters. Have the authors tried knocking down/out TP53 in a wild-type cell line to determine whether this is sufficient to enable cells to become persisters?

Cancer cell lines carrying wild type TP53, such as A375 and HCT116, do not show the persister phenotype. To examine the functional relevance of TP53, we stably expressed p53^{R273H} and p53^{R249S} mutants (Addgene #22934 and #22935) in A375 and HCT116, and indeed induced the persister phenotype. The results have been included in the revised manuscript as Fig S2G-H and shown below.

- Have the authors tested whether the chemo+mTORi persist phenotype they observe occurs *in vivo*? I recognize that this may be challenging to demonstrate, especially if chemotherapy alone *in vivo* is unable to completely eradicate the tumor cells. Alternatively, if chemotherapy alone *in vivo* does result in residual tumor cells and resistance, is this because *in vivo* conditions modify mTOR signaling in the tumor cells (even in the absence of mTORi) compared to *in vitro* conditions?

Thanks for the insightful comments. As Reviewer #3 pointed out, chemotherapy alone is rarely able to completely eradicate the tumors *in vivo*. Therefore, it is technically challenging to demonstrate the persist phenotype *in vivo*.

However, we examined the mTOR activity in a minimal residual tumor model based on the H441 xenografts treated with paclitaxel/cisplatin⁹. In the residual tumors following chemotherapy, we observed decreased mTOR activity as indicated by lower IHC staining using a phospho-p70 S6 Kinase (Ser371) antibody (PA5-38307, Invitrogen) (shown below). This observation suggests that even in the absence of mTOR inhibitors, the residual tumors show decreased mTOR activity *in vivo*.

- For Figure S4A, the IC50 values should also be shown to help interpret the data. Also, for the paclitaxel treatment, more doses are recommended to generate a more accurate dose response curve.

As our study focuses on the persister phenotype generated after long-term treatment, IC50 experiments determined on a 3-5 day timeline may not be as meaningful. Therefore, we have removed the IC50 profiles in the original Fig S4A.

MINOR COMMENTS:

- In Figure 1D, what are the units of the heatmap scale?, this is not labeled. Also, how were these genes ranked in the heatmap? This is not specified in the figure legend or methods.

The units for the heatmap scale are β scores generated from MAGeCK. The genes are ranked based on the average β scores of the two CRISPR screens. The top 15 enriched and 10 depleted genes are indicated in the figure. The figure and figure legend have been updated.

Additionally, I am curious to know what was the ranking of components of the mTORC2 complex in the CRISPR screen (Rictor, etc), as their data suggest the persister state is only dependent on mTORC1.

The ranking of mTORC2-specific components (Syap1, Mapkap1/mSin1, Prr5, Nckap1l, Prr5l, Rictor) and mTORC1 components (Telo2, Mtor, Mios, Raptor, Tsc2,Tsc1, Nprl2, Nprl3) is included in Fig S1A and shown below. In general, we do not see significant enrichment or depletion of mTORC2 components among the 19674 genes included in the CRISPR screens.

Gemcitabine			
	Gene	β score	Ranking
mTORC1	Telo2	3.4298	7
	Mtor	2.1908	71
	Rptor	1.4508	280
	Seh1l	1.4491	282
	Mios	1.2395	402
	Nprl3	-0.85581	19588
	Nprl2	-1.1221	19645
	Depdc5	-1.1451	19648
	Tsc1	-1.7931	19671
	Tsc2	-2.0205	19674
mTORC2	Syap1	0.17202	5499
	Mapkap1	0.070963	8589
	Prr5	-0.03719	12340
	Nckap1l	-0.15289	15910
	Prr5l	-0.26881	18039
	Rictor	-0.47997	19267

Selinexor			
	Gene	β score	Ranking
mTORC1	Mios	1.2968	197
	Telo2	1.2547	218
	Mtor	1.1296	282
	Rptor	0.77159	586
	Seh1l	0.55827	923
	Depdc5	-0.96665	19657
	Nprl3	-1.2883	19666
	Nprl2	-1.3555	19667
	Tsc1	-2.103	19672
	Tsc2	-2.5122	19673
mTORC2	Mapkap1	0.18162	2242
	Syap1	0.12043	2991
	Rictor	0.028486	7242
	Prr5l	-0.041884	14181
	Nckap1l	-0.070682	16516
	Prr5	-0.098164	17920

- How were the genes ranked in Figure S1? The details of ranking are not found in the figure legend or methods section.

The gene ranking was based on the average log₂ fold changes of seven CRISPR screens. We have updated the figure label and legend accordingly.

- For Figure 5A, the text mentions compounds targeting "cell death pathways (apoptosis and ferroptosis)", but these are not labeled as a distinct category in the figure.

We have updated the Fig 5A with the additional category of "cell death modulators".

- For Figure 6, the quantification of cell cycle based on the FUCCI reporter should be shown for the additional conditions in panel A (and not just for Gem + T)

The quantification data for all treatments have been updated in Fig 6B.

References

- 1 Vo, T. T. *et al.* mTORC1 Inhibition Induces Resistance to Methotrexate and 6-Mercaptopurine in Ph(+) and Ph-like B-ALL. *Mol Cancer Ther* **16**, 1942-1953, doi:10.1158/1535-7163.MCT-17-0024 (2017).
- 2 Fahy, L. *et al.* Hypoxia favors chemoresistance in T-ALL through an HIF1alpha-mediated mTORC1 inhibition loop. *Blood Adv* **5**, 513-526, doi:10.1182/bloodadvances.2020002832 (2021).
- 3 Dhir, T. *et al.* Abemaciclib Is Effective Against Pancreatic Cancer Cells and Synergizes with HuR and YAP1 Inhibition. *Mol Cancer Res* **17**, 2029-2041, doi:10.1158/1541-7786.MCR-19-0589 (2019).
- 4 Willabee, B. A. *et al.* Combined Blockade of MEK and CDK4/6 Pathways Induces Senescence to Improve Survival in Pancreatic Ductal Adenocarcinoma. *Mol Cancer Ther* **20**, 1246-1256, doi:10.1158/1535-7163.MCT-19-1043 (2021).
- 5 Leung-Pineda, V., Ryan, C. E. & Piwnicka-Worms, H. Phosphorylation of Chk1 by ATR is antagonized by a Chk1-regulated protein phosphatase 2A circuit. *Mol Cell Biol* **26**, 7529-7538, doi:10.1128/MCB.00447-06 (2006).
- 6 Borchering, N., Bormann, N. L., Voigt, A. P. & Zhang, W. TRGAted: A web tool for survival analysis using protein data in the Cancer Genome Atlas. *F1000Res* **7**, 1235, doi:10.12688/f1000research.15789.2 (2018).
- 7 van den Ende, T. *et al.* Neoadjuvant Chemoradiotherapy Combined with Atezolizumab for Resectable Esophageal Adenocarcinoma: A Single-arm Phase II Feasibility Trial (PERFECT). *Clin Cancer Res* **27**, 3351-3359, doi:10.1158/1078-0432.CCR-20-4443 (2021).
- 8 Snipstad, K. *et al.* New specific molecular targets for radio-chemotherapy of rectal cancer. *Mol Oncol* **4**, 52-64, doi:10.1016/j.molonc.2009.11.002 (2010).
- 9 Hegde, G. V. *et al.* Residual tumor cells that drive disease relapse after chemotherapy do not have enhanced tumor initiating capacity. *PLoS One* **7**, e45647, doi:10.1371/journal.pone.0045647 (2012).

REVIEWERS' COMMENTS

Reviewer #1 (Remarks to the Author):

The authors addressed most of the issues previously raised. The four panels that were provided in the rebuttal to answer my comments should be shown in the final publication, as these are important findings that sometimes differ from the published literature.

Reviewer #2 (Remarks to the Author):

The authors have addressed our previous concerns to our satisfaction. We have no further comments.

Reviewer #3 (Remarks to the Author):

The authors have addressed all of my comments and I am satisfied with the revisions.

Point-by-point Response to Reviewers' Comments

Reviewer #1 (Remarks to the Author):

The authors addressed most of the issues previously raised. The four panels that were provided in the rebuttal to answer my comments should be shown in the final publication, as these are important findings that sometimes differ from the published literature.

We thank Reviewer #1 for the comments and the enthusiasm on our study. We have included the single-cell tracking experiment as a supplementary figure in the final manuscript. Findings presented in the remaining three figures will be further developed and published in a separate study.

Reviewer #2 (Remarks to the Author):

The authors have addressed our previous concerns to our satisfaction. We have no further comments.

We thank Reviewer #2 for the efforts to review our revised manuscript.

Reviewer #3 (Remarks to the Author):

The authors have addressed all of my comments and I am satisfied with the revisions.

We thank Reviewer #3 for the efforts to review our revised manuscript.